# Context-enriched interactome powered by proteomics helps the identification of novel regulators of macrophage activation

Arda Halu[1,2], Jian-Guo Wang[2], Hiroshi Iwata[2], Alexander Mojcher[2], Ana Luisa Abib[2], Sasha A Singh[2], Masanori Aikawa[2]*, Amitabh Sharma[1†]*

[1]Channing Division of Network Medicine, Brigham and Women's Hospital, Harvard Medical School, Boston, United States; [2]Center for Interdisciplinary Cardiovascular Sciences, Brigham and Women's Hospital, Harvard Medical School, Boston, United States

**Abstract** The role of pro-inflammatory macrophage activation in cardiovascular disease (CVD) is a complex one amenable to network approaches. While an indispensible tool for elucidating the molecular underpinnings of complex diseases including CVD, the interactome is limited in its utility as it is not specific to any cell type, experimental condition or disease state. We introduced context-specificity to the interactome by combining it with co-abundance networks derived from unbiased proteomics measurements from activated macrophage-like cells. Each macrophage phenotype contributed to certain regions of the interactome. Using a network proximity-based prioritization method on the combined network, we predicted potential regulators of macrophage activation. Prediction performance significantly increased with the addition of co-abundance edges, and the prioritized candidates captured inflammation, immunity and CVD signatures. Integrating the novel network topology with transcriptomics and proteomics revealed top candidate drivers of inflammation. In vitro loss-of-function experiments demonstrated the regulatory role of these proteins in pro-inflammatory signaling.

*For correspondence:
maikawa@bwh.harvard.edu (MA);
amitabh.sharma@channing.
harvard.edu (AS)

†Deceased

**Competing interests:** The authors declare that no competing interests exist.

## Introduction

Pro-inflammatory macrophage activation plays a prominent role in a large number of disorders including cardiovascular disease (CVD) (*Aikawa and Libby, 2004*; *Glass and Olefsky, 2012*; *Glass and Witztum, 2001*; *Gregor and Hotamisligil, 2011*; *Liang et al., 2007*; *Randolph, 2014*; *Ridker and Lüscher, 2014*; *Tabas, 2010*). Established treatments for CVD such as those dependent on the cholesterol lowering effect of statins do not completely eliminate cardiovascular risk (*Aikawa and Libby, 2004*; *Aikawa et al., 2001*; *Libby, 2005*), therefore alternative novel solutions are needed to tackle such residual risk by targeting pro-inflammatory activation in CVD (*Ridker et al., 2017*). Characterizing the mechanisms underlying macrophage activation itself proves to be a challenging task, given the functional heterogeneity of macrophages and the complex interplay between the pro- and anti-inflammatory phenotypes (*Biswas and Mantovani, 2012*; *Gordon and Mantovani, 2011*; *Koltsova et al., 2013*; *Lawrence and Natoli, 2011*; *Ley et al., 2011*; *Moore et al., 2013*; *Murray et al., 2014*; *Swirski and Nahrendorf, 2013*). Furthermore, it is increasingly recognized that macrophage activation has many distinct types and follows a spectrum model defined by specific stimuli rather than the bipolar model of pro- and anti-inflammatory polarization that once prevailed (*Murray et al., 2014*; *Nahrendorf and Swirski, 2016*; *Xue et al., 2014*). Nevertheless, using experimental models, where cause-effect relations are well defined, within a

**eLife digest** When human cells or tissues are injured, the body triggers a response known as inflammation to repair the damage and protect itself from further harm. However, if the same issue keeps recurring, the tissues become inflamed for longer periods of time, which may ultimately lead to health problems. This is what could be happening in cardiovascular diseases, where long-term inflammation could damage the heart and blood vessels.

Many different proteins interact with each other to control inflammation; gaining an insight into the nature of these interactions could help to pinpoint the role of each molecular actor. Researchers have used a combination of unbiased, large-scale experimental and computational approaches to develop the interactome, a map of the known interactions between all proteins in humans. However, interactions between proteins can change between cell types, or during disease. Here, Halu et al. aimed to refine the human interactome and identify new proteins involved in inflammation, especially in the context of cardiovascular disease.

Cells called macrophages produce signals that trigger inflammation whey they detect damage in other cells or tissues. The experiments used a technique called proteomics to measure the amounts of all the proteins in human macrophages. Combining these data with the human interactome made it possible to predict new links between proteins known to have a role in inflammation and other proteins in the interactome. Further analysis using other sets of data from macrophages helped identify two new candidate proteins – GBP1 and WARS – that may promote inflammation. Halu et al. then used a genetic approach to deactivate the genes and decrease the levels of these two proteins in macrophages, which caused the signals that encourage inflammation to drop.

These findings suggest that GBP1 and WARS regulate the activity of macrophages to promote inflammation. The two proteins could therefore be used as drug targets to treat cardiovascular diseases and other disorders linked to inflammation, but further studies will be needed to precisely dissect how GBP1 and WARS work in humans.

systems-based approach might help to facilitate the discovery of specific mechanisms that can contribute to the overall balance of macrophage phenotype or new therapeutic targets. As it stands, the hunt for hitherto undiscovered mechanistic connections in macrophage activation, and therapeutic targets aimed at resultant CVD, has much to benefit from complex systems approaches emerging in medicine.

Complex human diseases such as CVD are seldom the result of a perturbation of a single gene but rather arise from the concerted effects of multiple genes and their products forming complex networks of interactions in cells, collectively embodied in the 'interactome' concept (*Vidal et al., 2011*). Network medicine has emerged as an effective quantitative framework to address the complexity of human disease (*Barabási et al., 2011*). Increasing evidence suggests that disease determinants tend to be localized in the same region in the interactome and interact closely with each other, leading to their organization into 'disease modules' (*Barabási et al., 2011*; *Goh et al., 2007*; *Menche et al., 2015*). The same principle is valid for functionally similar genes, which form 'functional modules' (*Shih and Parthasarathy, 2012*). Based on this understanding of biological function and disease pathogenesis, network-based methods have successfully identified candidates for therapeutic targets: Early studies that constructed the network of approved drug targets and disease genes shed light on the current trends in drug discovery (*Yildirim et al., 2007*). Other network-based pharmacological studies have focused on drug target prediction and proposed drug repurposing methods (*Berger and Iyengar, 2009*; *Wu et al., 2013*). Methods using shortest paths between drug targets (*Gottlieb et al., 2011*; *Guney et al., 2014*; *Lee et al., 2012*; *Zhao and Li, 2012*) and drug-disease proximity (*Guney et al., 2016*) in the interactome were proposed.

Despite these advances, one important factor hampering the effective molecular characterization of diseases is that interactomes, which are collections of multiple types of literature-curated physical protein-protein interactions (PPIs) with experimental evidence from high-throughput experiments and small-scale studies, are incomplete (*Menche et al., 2015*) and biased toward highly studied proteins such as disease-related proteins or targets of pre-existing drugs (*Rolland et al., 2014*). Most importantly, interactomes are context-independent, that is do not carry information specific to the

cell type, experimental condition or pathological state but rather represent the sum of all observed interactions, which makes them inherently generic and limits their utility in diverse biological and experimental settings.

Owing to its unbiased nature and high information content, high-resolution/accuracy mass spectrometry (MS) offers an opportunity to characterize the proteome of a specific cell state in a comprehensive way (*Mann et al., 2013*). The rapid progress in proteomics technologies has instigated a growing body of works that combine large amounts of MS data with network biology methods. For instance, a recent study has used the proteomics of breast cancer cells in conjunction with literature-derived signaling networks (*Sacco et al., 2016*). Protein abundance profiles from global proteomics measurements have been used to cluster co-regulated proteins (*Singh et al., 2014*), as well as build 'co-abundance' networks to identify key driver proteins for viral replication (*McDermott et al., 2012*). Thus, cell- and tissue-specific proteomic profiling could complement the incomplete and generic yet system-wide cellular picture provided by the interactome, especially in specific disease contexts.

Here, we used unbiased macrophage-derived proteomics measurements to enhance the literature-curated human interactome by adding cell type- and condition-specific information to it, thereby helping address its context-independence as well as its incompleteness. While we now realize that macrophage heterogeneity is more multidimensional rather than M1/M2 dichotomy (*Murray et al., 2014*), assessing molecular mechanisms still requires a model in which cause-effect relationships are well defined, as we recently demonstrated (*Iwata et al., 2016*). We thus used interferon-γ (IFNγ) as an example of major instigators of pro-inflammatory macrophage activation. We utilized a network proximity-based prediction method to identify key drivers of macrophage activation as it pertains to CVD. Our results revealed that edges derived from macrophage-specific proteomics contributed to the less characterized parts of the interactome, reflected the respective macrophage stimulation condition in terms of pathways and biological processes, and increased the prediction performance of CVD therapeutic targets. The top-ranked candidates for regulators of macrophage activation, and hence potential CVD drug targets, also showed significant enrichment with immune system as well as cardiovascular disease related signatures. Our multi-step and multi-omics analytical pipeline resulted in the identification of Guanylate binding protein 1 (GBP1) and tryptophanyl-tRNA synthetase (WARS) as top candidates, based on evidence from network topology, gene expression and protein expression. To validate our network-based prediction, we performed loss-of-function experiments and demonstrated that GBP1 and WARS indeed regulate the expression of the pro-inflammatory cytokine, CCL2, and phosphorylation of STAT1, two classical pro-inflammatory readouts. Overall, our findings suggest the utility of adding context-specific information to the generic interactome.

## Results

### Co-abundance networks from a cell culture model of macrophage activation add specificity to the interactome while recapitulating known interactions

The literature-curated human interactome, or PPI network (see Materials and methods), hosts invaluable information about potential protein subnetworks related to diseases. However, it is non-specific as it describes interactions that can occur between proteins within any cell or tissue under any condition, as well as currently incomplete. We hypothesized that the introduction of phenotype-specific interactions to the PPI network would address its incompleteness and fill in the biologically less characterized portions of it. We used time-course proteomics data collected from baseline control, IFNγ stimulated/pro-inflammatory, and IL-4 stimulated anti-inflammatory/pro-resolving phenotypes of the macrophage-like cell line THP-1, denoted by M(-), M(IFNγ) and M(IL-4), respectively, as the phenotypic information.

For each condition, we built co-abundance networks to represent the relationships between proteins that show similar abundance patterns over time by constructing the correlation matrix, setting a correlation threshold and building the network based on the edges above this threshold (*Figure 1A* and *Figure 1—figure supplement 1A*, see Materials and methods and *Supplementary file 1* for a summary of topological properties). While the co-abundance networks

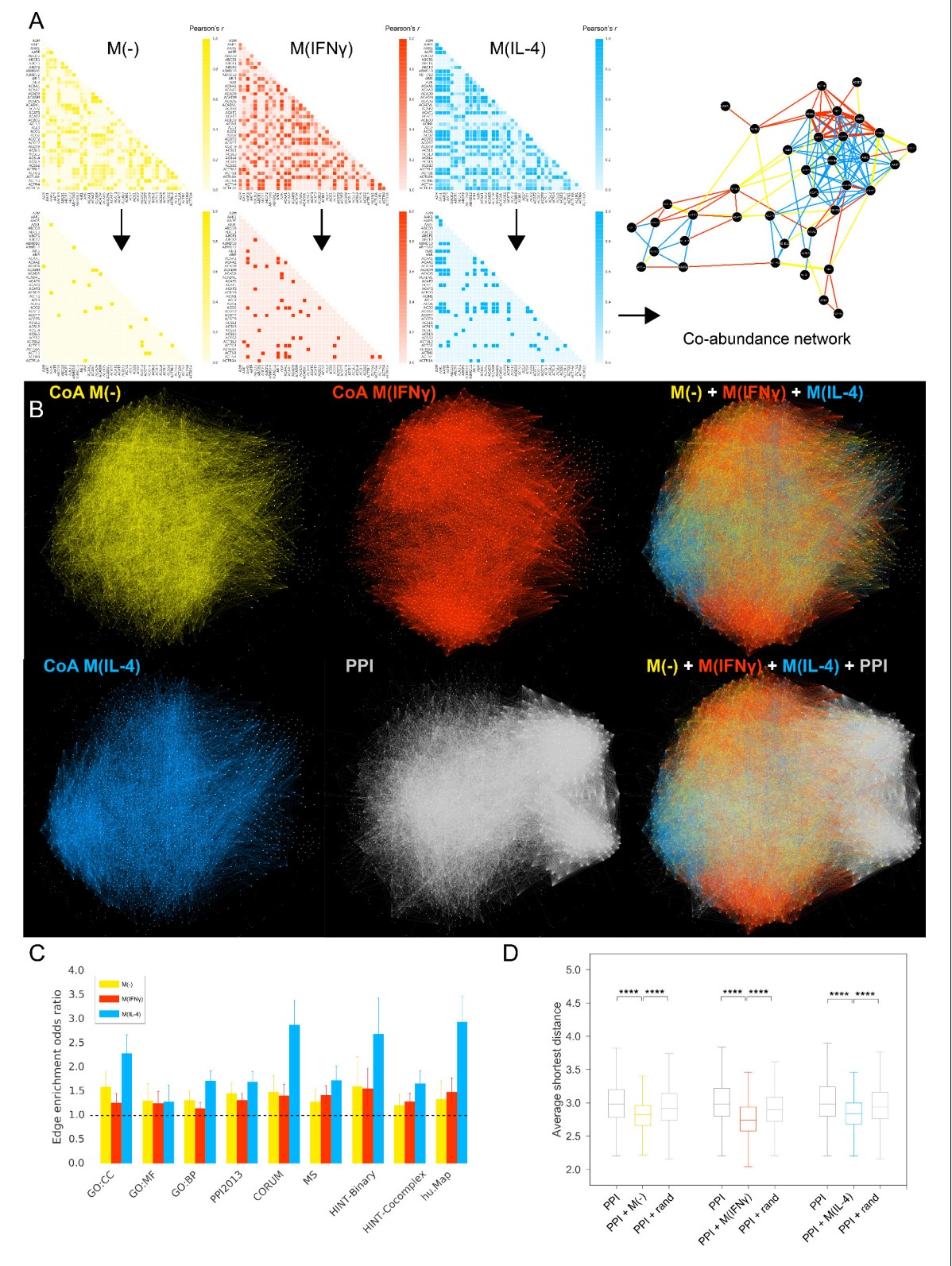

**Figure 1.** Co-abundance networks. (**A**) Top row: Sample from the Pearson correlation matrix showing the top 40 proteins in alphabetical order. Pearson's r values are calculated based on the protein abundance profiles for each condition. Bottom row: Correlation matrices after thresholding, where Pearson's r values above the defined threshold are assigned the value 1 and r values below the threshold are assigned the value 0, resulting in the adjacency matrix for the co-abundance network. Right: The resulting co-abundance network. Yellow, red and blue edges correspond to co-

*Figure 1 continued on next page*

*Figure 1 continued*

abundance edges derived from proteomics data from the M(-), M(IFNγ) and M(IL-4) condition, respectively. (B) The global superimposition of co-abundance networks and the literature-derived PPI network, where the same force-directed network layout was used, preserving the spatial positions of nodes. The depicted PPI network was pruned to contain only the proteins in the co-abundance networks. (C) The enrichment of co-abundance edges in external validation datasets of functional, binary and co-complex interactions including shared GO terms for Cellular Component (GO:CC), Molecular Function (GO:MF) and Biological Process (GO:BP), binary interactions from the HINT database (HINT-Binary), the literature-curated PPI network used in the subsequent analysis (PPI2013), and co-complex interactions from large-scale databases (CORUM, HINT Co-complex, hu.Map) and mass-spectrometry-based curated co-complex association maps (MS). Enrichments were calculated using two-sided Fisher's Exact test. Error bars correspond to 95% confidence intervals. All enrichments were significant with p-values<0.05. (D) Distributions of average shortest distances between the co-abundance network nodes and known CVD drug targets. Random edge addition was done for 100 realizations using degree-preserving randomization (see Materials and methods). ****p<0.0001, two-sided Mann-Whitney U test with Bonferroni correction for multiple testing.

The online version of this article includes the following source data and figure supplement(s) for figure 1:

**Source data 1.** Source data for *Figure 1A*: Co-abundance networks in edgelist format (Columns: Protein A - Protein B).
**Source data 2.** Source data for *Figure 1A*: Co-abundance networks in edgelist format (Columns: Protein A - Protein B).
**Source data 3.** Source data for *Figure 1A*: Co-abundance networks in edgelist format (Columns: Protein A - Protein B).
**Source data 4.** Source data for *Figure 1C*: Odds ratios and confidence intervals, and p-values for the link validation sets shown in *Figure 1C*.
**Figure supplement 1.**

have several-fold higher edge densities than the PPI network as a whole (*Supplementary file 1*), their edge densities are comparable to the region of the PPI that hosts the proteins in the co-abundance networks (0.96%). Mirroring well-known properties of PPI networks, the co-abundance networks display broad degree distributions (*Figure 1—figure supplement 1B*) and high average clustering coefficients (*Supplementary file 1*). Despite these similar characteristics, in comparison with the PPI network, the co-abundance networks have higher diameters and higher average shortest path lengths, as well as higher average clustering coefficients, possibly due to similar protein expression patterns connecting local groups of proteins all at once (*Supplementary file 1*). Thus, co-abundance networks are more locally clustered and not as compact, compared to the PPI network, overall suggesting a complementary topology to the PPI network.

We then asked what the topologies of co-abundance networks add to the PPI network. Overlaying the co-abundance networks on the PPI network, we found that each stimulation condition was denser in certain regions, and overall filled in the sparse parts of the PPI network, resulting in a combined interactome where each condition can be distinguished and their overlapping regions can be assessed (*Figure 1B*). Thus, co-abundance networks may be instrumental in helping address the lack of biological investigation of certain parts, that is the incompleteness, of the PPI network while enhancing it with experiment- and context-specific information.

To ensure that the co-abundance edges correspond to biologically meaningful interactions, we validated these edges by measuring their overlap with known physical interactions and shared functional annotations. Co-abundance edges showed significant enrichment for (i) shared Gene Ontology (GO) terms, (ii) the literature curated PPI network used throughout this study, and (iii) interactions between pairs of proteins belonging to a common protein complex, based on high-quality protein complex databases and curated co-complex association maps (*Figure 1C*). Overlaps with large-scale binary and affinity-purification-mass spectrometry (AP-MS) based interactomes were not significant (*Supplementary file 2*), possibly due to the strict experimental criteria for these maps, while the 'binary' portion of the literature-curated HINT database (*Das and Yu, 2012*) showed significant enrichment (*Figure 1C*, see Materials and methods for details on the databases used). Thus, the co-abundance networks independently generated from MS data in an experiment-specific context still capture GO-term-based functional associations and high-confidence interactions documented in context-independent interactomes.

## Addition of macrophage derived co-abundance edges shortens the paths to CVD drug targets

We next inquired about how the macrophage-specific topology provided by co-abundance edges could facilitate the extraction of CVD and inflammation related drug target information from the combined network. To test this, we evaluated the effect of the addition of co-abundance edges on the shortest paths between proteins and drug targets related to the biology of the co-abundance

network, and hence the combined network's overall 'drug target navigability.' We found that, for both M(IFNγ) and M(IL-4), the average shortest distances between the co-abundance network's nodes and known CVD drug targets from the integrated Complex Traits Network (iCTNet) database (*Wang et al., 2015*) (see Materials and methods, *Figure 1—figure supplement 1C*) are shorter for the combined network than the PPI network only (*Figure 1D*). Proving that this effect cannot be attributed to the mere addition of edges, the addition of co-abundance edges resulted in an average shortest distance distribution that is shifted significantly towards lower values compared to the addition of the same number of randomly chosen edges (*Figure 1D*). Overall, these findings indicate that the co-abundance links used in conjunction with the PPI network result in a more compact network structure that makes the known drug targets more reachable by other proteins.

## Addition of macrophage derived co-abundance edges increases CVD drug target prediction performance

We hypothesized that, since the macrophage-derived co-abundance network renders the PPI network more compact and navigable to drug targets, network proximity-based drug target prediction methods should fare better on this combined network than the PPI network alone, enabling us to more efficiently identify novel targets. The co-abundance networks spanned a certain portion of the entire PPI network with their edges concentrated on certain regions, while the CVD drug targets were dispersed around the combined PPI network, forming a disconnected subnetwork (*Figure 2A* and *Figure 2—figure supplement 1A*), suggesting the fragmented nature of the CVD drug target information in the interactome. The proportions of CVD drug targets to non-drug targets in the PPI and co-abundance network were very similar to each other (0.0018 versus 0.0017, respectively), suggesting that there is no preferential presence of drug targets in either network over the other one. In terms of connectivity, the degrees of drug targets were significantly higher than the degrees of other proteins in the PPI network, while there was no significant degree difference between drug targets and other proteins in the co-abundance networks (*Figure 2—figure supplement 1B*). Furthermore, measuring the tendency of co-abundance edges to connect to hubs in the PPI network, we found that proteins connected by CoA edges (i.e. CoA network nodes) have significantly higher degrees in the PPI network than the other nodes in the PPI network (*Figure 2—figure supplement 1C*). Taken together, these results point at a degree bias in the way drug targets and co-abundance edges are connected in the PPI network. To give equal opportunity to proteins not necessarily captured by proteomics measurements but contained in the PPI network, we chose to leverage the biology-specific information provided by the co-abundance network for a global prioritization of all proteins in the combined PPI network.

We utilized a drug target prioritization method inspired by similar methods based on the network proximity between candidates and seeds (*Guney et al., 2016*; *Krauthammer et al., 2004*), which uses a metric that is a function of the average shortest distance between the candidate and the set of known CVD drug targets (seeds) from iCTNet (*Figure 2A*, see Materials and methods). As a proxy of prediction performance, we measured the rate of detecting known CVD drug targets by calculating the area under the receiver operating characteristic (ROC) curve. Compared to the PPI network only, the area under the ROC curve (AUROC) was significantly increased with the addition of co-abundance links (p=0.018, p=0.003 and p=0.015 for M(-), M(IFNγ) and M(IL-4), respectively, paired t-test following k-fold cross-validation) (*Figure 2B*, *Figure 2—figure supplement 2A–C*). The prediction performance of the combined networks also surpassed those of co-abundance networks alone (*Figure 2—figure supplement 3A*). The distribution of known CVD drug targets ordered by prediction rank indeed showed that they were more predominantly ranked in higher quantiles in the three co-abundance enriched networks compared to the PPI network only (*Figure 2C*). Furthermore, we found that the prediction performance can be further improved by (a) the systematic 'de-noising' of the PPI network by removing edges based on low-throughput experiments and co-complex interactions (see Materials and methods, and *Figure 2—figure supplement 3B–D*) and (b) modulating the ratio of weights of the PPI links and co-abundance links, specifically by giving more 'importance' to the co-abundance links (see Materials and methods and *Figure 2—figure supplement 4A–B*), hence increasing the specificity of the network to the biological question.

Finally, to ensure that the increase in the prediction performance is not simply the result of the bulk addition of edges, we compared the AUROCs of co-abundance enriched PPI networks to those enriched with the equivalent number of random edges. To control for the degree bias in the

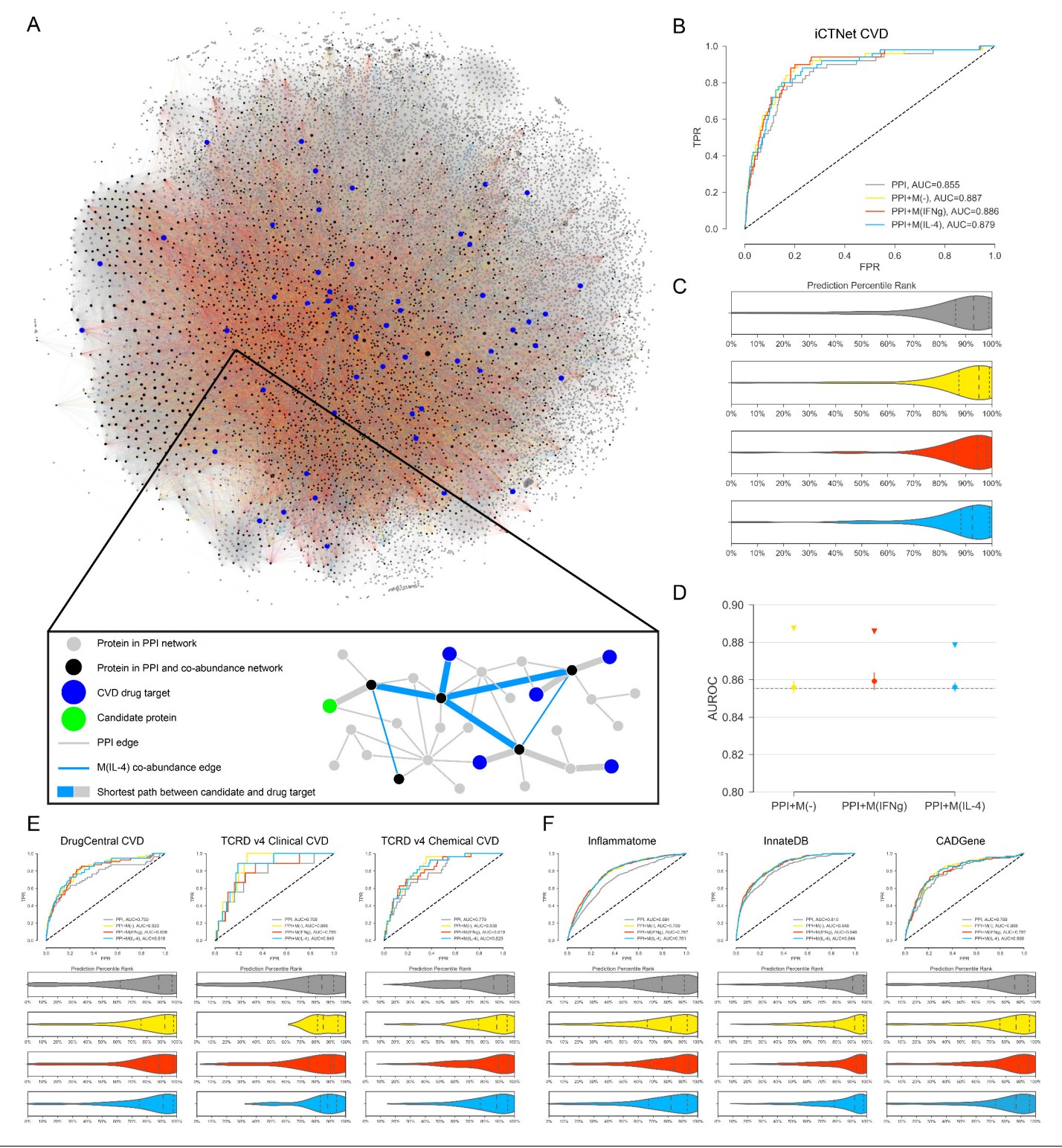

**Figure 2.** Target candidate prioritization and its performance assessment. (**A**) The entire literature curated PPI network with co-abundance edges from all three stimulation conditions, providing a global view of the distribution and connectivity of co-abundance edges and drug targets (*Figure 2—figure supplement 1* for a quantification of related degree distributions and largest connected component (LCC) sizes). A force-directed layout algorithm was used to visualize the networks. Edge colors follow the same convention as *Figure 1*. (Inset) Toy network depicting the drug target prioritization scheme: For each candidate protein (green node), the shortest path length to each CVD drug target (blue nodes) is calculated and the proximity score $PS(c)$ is calculated (see Materials and methods). Shortest paths between the candidate and CVD drug targets are denoted with the thicker edges and may

*Figure 2 continued on next page*

*Figure 2 continued*

consist of both PPI and co-abundance edges. (B) The ROC curves for iCTNet CVD drug targets on PPI network only and PPI network with added co-abundance edges, with AUROCs shown in the legend. (C) Violin plots showing the distribution of percentile ranks of iCTNet CVD drug targets on PPI network only and PPI network with added co-abundance edges. Dashed lines indicate the 2nd quartiles (medians) and dotted lines indicate the 1st and 3rd quartiles. (D) AUROC value comparison between PPI and co-abundance edges (indicated by triangles) and PPI and the same number of randomly added edges as the co-abundance networks (circles with error bars). Randomization was repeated for 20 realizations using degree-preserving randomization (see Materials and methods). All empirical p-values between co-abundance and random case are less than 0.05. The AUROC of the PPI network is indicated by the grey dashed line. (E) ROC curves and prediction percentile rank violin plots for external drug target databases: DrugCentral, TCRD Clinical and TCRD Chemical. Violin plots show the distribution of percentile ranks of CVD drug targets from these databases for each case. Dashed lines indicate the 2nd quartiles (medians) and dotted lines indicate the 1st and 3rd quartiles. (F) ROC curves and prediction percentile rank violin plots for inflammation (Inflammatome), innate immune response (InnateDB) and coronary artery disease (CADGene) signatures. Violin plots show the distribution of percentile ranks of proteins implicated in these datasets for each case. Dashed lines indicate the 2nd quartiles (medians) and dotted lines indicate the 1st and 3rd quartiles.

The online version of this article includes the following figure supplement(s) for figure 2:

**Figure supplement 1.**
**Figure supplement 2.** For the iCTNet CVD drug targets.
**Figure supplement 3.** For the iCTNet CVD drug targets.
**Figure supplement 4.**
**Figure supplement 5.**
**Figure supplement 6.**
**Figure supplement 7.**
**Figure supplement 8.**
**Figure supplement 9.**
**Figure supplement 10.**

connectivity between co-abundance and PPI networks (*Figure 2—figure supplement 1B–C*), we implemented degree-preserving randomization, which ensures the addition of randomly selected edges that connect proteins with similar degrees to the proteins in the co-abundance networks (see Materials and methods). We found that co-abundance networks contribute to the increase in prediction performance significantly more (empirical p-value<0.05) than the random case (*Figure 2D*).

To establish the relevance of our network-based prioritization to the therapeutic targets it is aimed at finding, we sought the enrichment of the top-ranked proteins in external datasets. As iCTNet-based CVD targets were used as seeds in the prioritization scheme, we used external datasets containing clinically approved CVD targets and chemically suitable CVD targets satisfying given small molecule activity thresholds for validation (see Materials and methods). These datasets were fairly orthogonal, showing insignificant overlap (except for one case, where the overlapping genes were removed from both datasets, see *Figure 2—figure supplement 4C*) between their CVD drug target sets, thereby providing three independent sources for validation (*Figure 2—figure supplement 4C*). We once again observed the increase in the target prediction performance with the addition of co-abundance edges (*Figure 2E*, *Figure 2—figure supplement 5*). These findings applied to all drug targets derived from the same databases as well (*Figure 2—figure supplements 6* and *7*), reflecting the broad impact of inflammation in human disease and suggesting the potential of macrophage-specific co-abundance edges in capturing additional inflammatory disease drug targets.

We also tested how robust the prediction performance is against changes in several points in the workflow. First, to investigate whether the improvement in the prediction performance is influenced by potential shared dependencies on a third confounding factor, we implemented two correlation measures that are robust against outliers and control for confounding factors, biweight midcorrelation and partial correlation, to build the co-abundance networks (see Materials and methods). The improvement in the AUROCs with the addition of co-abundance networks over PPI only was similar to the Pearson coefficient case (*Figure 2—figure supplement 8*). These results indicate that the outlier abundance values and the indirect effect of baseline abundances do not influence the downstream analysis substantially.

Second, to explore how sensitive prediction performance is to untraversed longer-range paths between candidates and drug targets, we have considered global association measures and other types of network-based distance measures. In particular, we used dynamical prioritization methods such as random walk with restarts (RWR) (*Köhler et al., 2008*) and its degree-aware version (DADA)

(*Erten et al., 2011*), as well as other distance-based measures such as the kernel distance (*Guney et al., 2016*). Relying on a random walker, RWR/DADA takes into account all possible paths, including many longer-range paths between a candidate and a target (seed), whereas Kernel distance penalizes paths based on their length using an exponential function (see Materials and methods). Both types of measures yielded similar results to average shortest path in terms of the AUROC, and the improvement with the addition of CoA edges persisted (*Figure 2— figure supplement 9*).

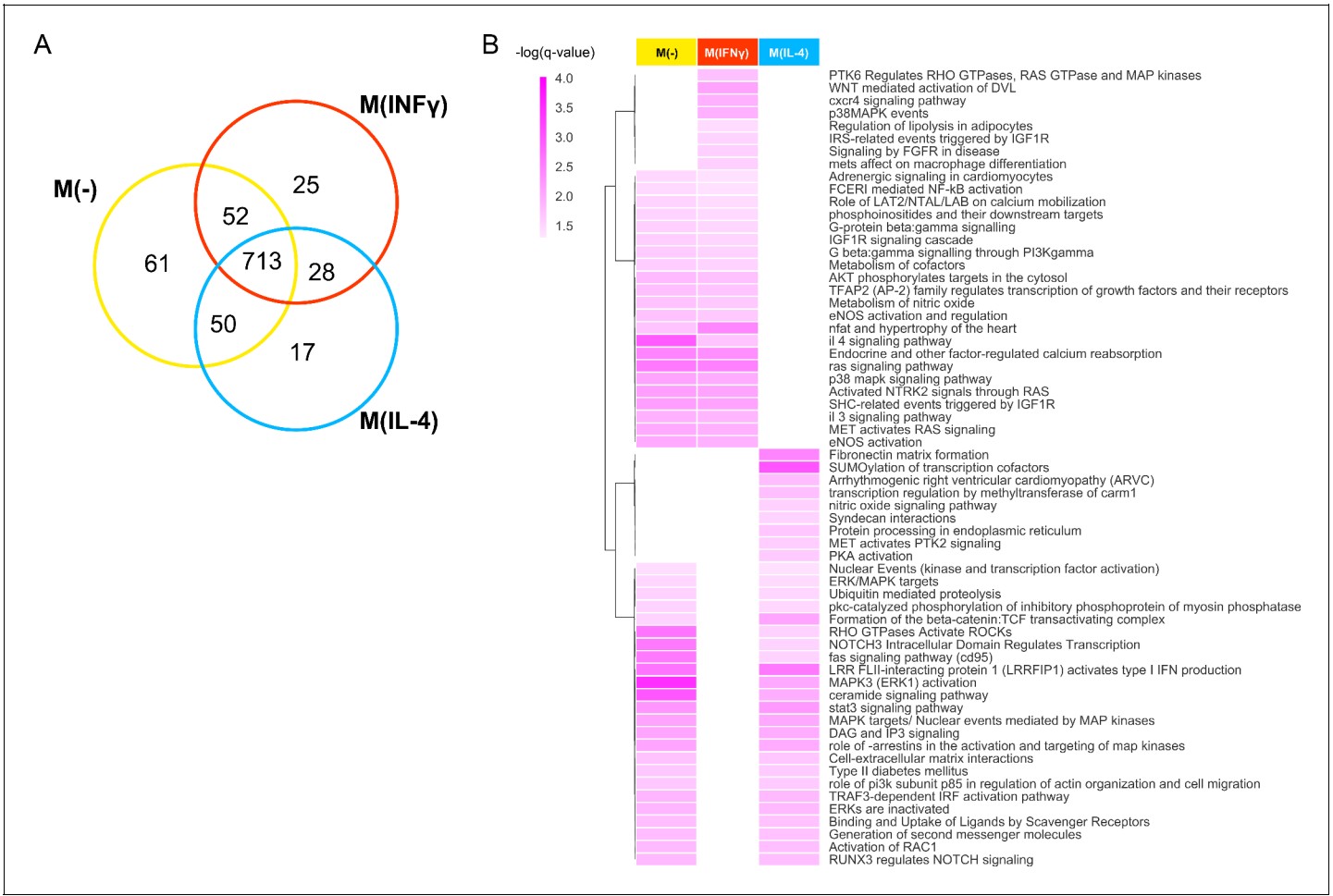

**Figure 3.** Pathways of top prioritized proteins. (**A**) The Venn diagram showing the shared pathways between the significantly enriched (q-value <0.05) pathways of the top candidates (empirical p-value<0.01) prioritized by the addition of the M(-), M(IFNγ) and M(IL-4) co-abundance networks to the PPI network. (**B**) The hierarchically clustered condition-specific pathways (see *Figure 3—figure supplement 1* for a full list). Darker shades of colors indicate a higher enrichment in terms of the negative logarithm of q-value.

The online version of this article includes the following source data and figure supplement(s) for figure 3:

**Source data 1.** Source data for *Figure 3*: All enriched pathways of the top N (p<0.01) prioritized proteins.

**Figure supplement 1.** The full list of hierarchically clustered condition-specific pathways.

**Figure supplement 1—source data 1.** The enrichment p-values of the condition-specific pathways shown in the clustered heatmap of *Figure 3—figure supplement 1*.

**Figure supplement 2.** The network of significantly enriched pathways of the top-ranked (empirical p-value<0.01) candidates prioritized by the addition of the M(IFNγ) co-abundance network to the PPI network.

**Figure supplement 3.** The network of significantly enriched pathways of the top-ranked (empirical p-value<0.01) candidates prioritized by the addition of the M(IL-4) co-abundance network to the PPI network.

## Top prioritized candidates capture immunity and CVD signatures and reflect the respective macrophage phenotype

To ensure that the prioritization method above reflects the macrophage biology and cardiovascular disease relation of the co-abundance networks, we measured their rate of capturing genes related to inflammation, innate immune response, and coronary artery disease signatures (see Materials and methods). The performance of the co-abundance enriched PPI networks in capturing these signatures surpassed that of the PPI network alone (*Figure 2F*, *Figure 2—figure supplement 10*). The top prioritized (empirical p-value<0.01, see Materials and methods) were significantly enriched in all three datasets for both stimulations (*Supplementary file 3*).

The majority of the enriched pathways of the top prioritized proteins (*Figure 3—figure supplement 1—source data 1*) were common (*Figure 3A*). This is mainly due to the fact that the despite the different stimulation conditions, the biological processes and pathways of top-prioritized candidates are mainly dominated by the cell type, resulting in a large number of commonalities, and reflecting the predominant effect of the common macrophage biology on the resulting target prioritization. Focusing on the condition-specific pathways, we found that M(IFNγ) is enriched in pro-inflammatory pathways such as p38 MAPK and NF-κB activation related pathways, as well as pathways related to PI3K-Akt activation, NFAT and hypertrophy of the heart, adrenergic signaling in cardiomyocytes and interleukin signaling, whereas M(IL-4) is enriched in Rho GTPase activation, SUMOylation, β-catenin, scavenger receptor and Fas (CD95) signaling related pathways (*Figure 3B*, see *Figure 3—figure supplement 1* for full list). Mapping these pathways in a pathway network (see Materials and methods) enabled us to summarize all biological processes related to each condition whereby pathways sharing molecular elements were clustered together (*Figure 3—figure supplements 2* and *3*). Together, these results demonstrate that our global ranking of the entire PPI network, including proteins not in macrophage proteomics, captures the inflammatory and immune-response-related component of CVD mediated by macrophage activation.

## Combined ranking based on network topology, gene expression and protein abundance reveals novel regulators of macrophage activation

Combining proteomic and transcriptional information improves the identification of key driver molecules (*Padi and Quackenbush, 2015*). In a similar vein, we sought to integrate our protein abundance data with previously published gene expression data from human macrophages (*Xue et al., 2014*) to refine our list of prioritized proteins and focus on a much smaller subspace of highly expressed candidates for further in vitro validation. Particularly, since the network-based prioritization ranks all the proteins in the interactome indiscriminately and the prediction ROC curves show the greatest improvement at moderate ranks (*Figure 2E*), we devised a combined filtering/ranking scheme to obtain a smaller set of final drug target candidates while maximizing the advantage offered by the performance of our prioritization method. The filtering step was used to intersect the network-based prioritization ranking with the highly expressed molecules from -omics data to detect the strongest signals that were also close to drug targets. The ranking scheme considered (a) the network closeness to drug targets, (b) the relative protein abundance difference with respect to baseline M(-) over all time points, and (c) the gene expression fold change with respect to baseline M(-) from human macrophage transcriptome data, and calculated a combined score based on these three criteria (*Figure 4A* and *Figure 4—figure supplement 1*, see Materials and methods).

To quantify each constituent ranking's relative contribution to the final ranking, we investigated the correlations between them. The combined ranking was positively correlated with all separate rankings. While the combined ranking was slightly more driven by the gene expression and protein abundance, the network prioritization ranking was close to them for M(IFNγ) and on par with them for M(IL-4) (*Figure 4—figure supplement 2A–B*). Moreover, each separate ranking was orthogonal to each other with insignificant correlations (*Figure 4—figure supplement 3A–B*). Together, these results suggest that network-based prioritization ranking contributes in a non-trivial way to the combined ranking, and that each ranking carries its own information, contributing uniquely to the combined ranking.

The final list of candidates ranked according to the combined score was significantly close in the interactome to CVD drug targets for both M(IFNγ) and M(IL-4) in terms of the average shortest distance than what would be expected by chance (*Figure 4B*, see Materials and methods), confirming

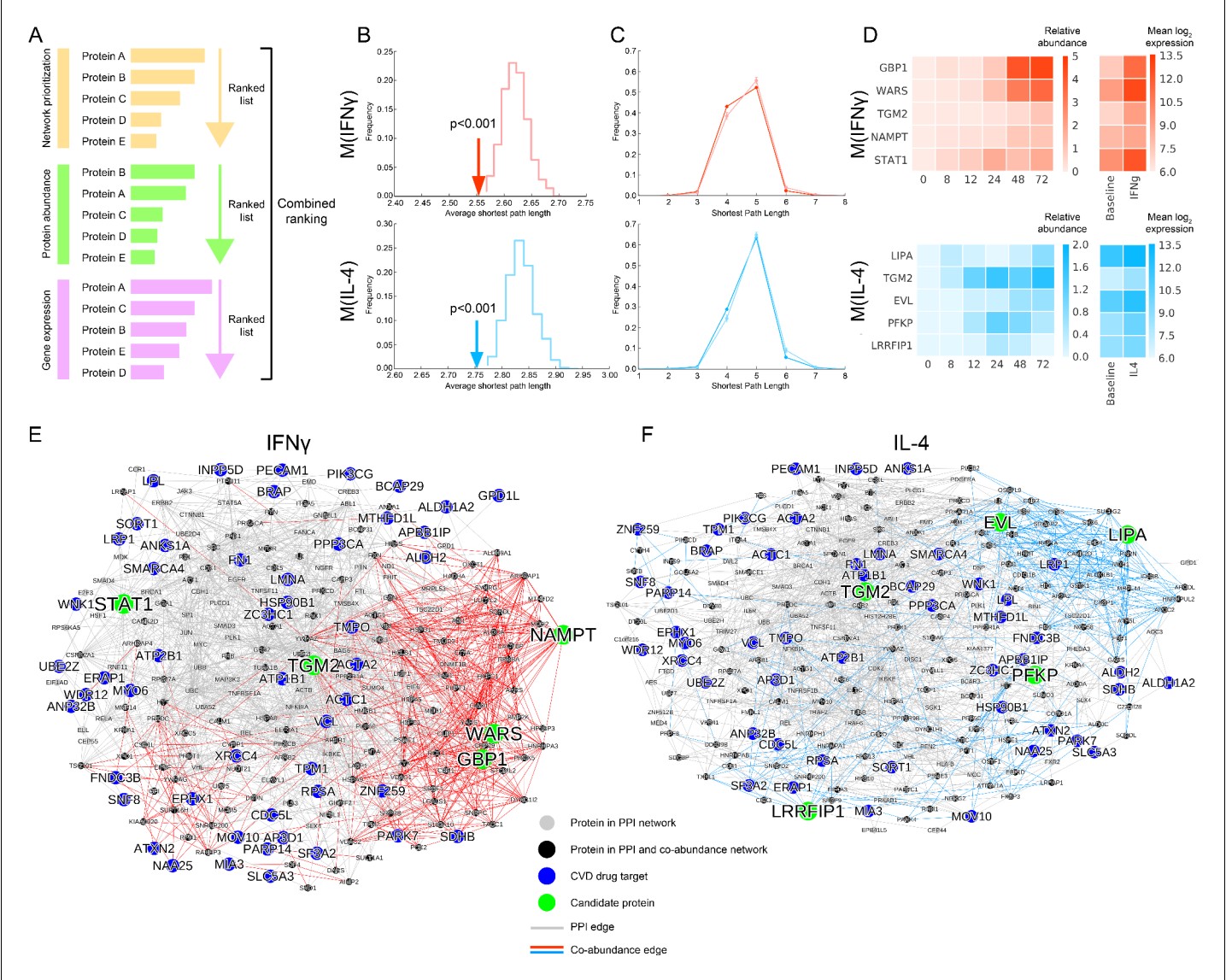

**Figure 4.** Selection of top target candidates by incorporating expression and abundance data. (**A**) Overview of the filtering/combined ranking scheme. Three rankings were performed according to: (i) the network prioritization, (ii) the relative protein abundance difference with respect to baseline M(-) over all time points, and (iii) the gene expression fold change with respect to baseline M(-) from human macrophage transcriptome data (*Xue et al., 2014*) (see Mateials and methods). (**B**) The average shortest path length between the filtered candidates and CVD drug targets (shown with arrows) compared to random expectation (shown as the distribution), with red indicating M(IFNγ) and blue indicating M(IL-4). Degree-preserving node randomization was performed for 1000 realizations. (**C**) The shortest path length distributions between the filtered candidates and CVD drug targets. Darker shades indicate the real shortest path length distribution and the lighter shades with error bars indicate random expectation. Degree-preserving node randomization was performed for 1000 realizations. Shorter path lengths (such as four steps) are significantly over-represented in the real case compared to random, whereas longer path lengths (such as six steps) are significantly under-represented in the real case compared to random (empirical p-values<0.001). (**D**) Relative protein abundance and gene expression (mean log 2 transformed) values of the top five ranked candidates. The relative protein abundance was calculated by subtracting the baseline M(-) for all time points between and including 0 and 72 hr. (**E**) Subnetwork showing the paths between the top five ranked candidates from M(IFNγ) and CVD drug targets. Grey edges indicate PPI network edges and red edges indicate M(IFNγ) co-abundance edges. (**F**) Subnetwork showing the paths between the top five ranked candidates from M(IL-4) and CVD drug targets. Grey edges indicate PPI network edges and blue edges indicate M(IL-4) co-abundance edges. In (**E**) and (**F**), the networks were constructed by calculating all shortest paths between every (top candidate, CVD target) pair. Node size corresponds to node degree.

The online version of this article includes the following figure supplement(s) for figure 4:

**Figure supplement 1.**
**Figure supplement 2.**
**Figure supplement 3.**

that the filtering of candidates aligns with the original premise of network proximity to CVD drug targets. This finding was further supported in the discretized distribution of shortest path lengths (*Figure 4C*, see Materials and methods), where shorter path lengths were significantly over-represented compared to random expectation. The top five final candidates for the M(IFNγ) ranking were GBP1, WARS, TGM2, NAMPT and STAT1, excluding PARP14, which was among the iCTNet drug targets. STAT1 belongs to a family of molecules known to be expressed in the heart (*Xuan et al., 2001*), and to play a role in the link between coronary artery disease and inflammatory responses in vascular cells (*O'Donnell and Nabel, 2011*). Of these candidates, GBP1 and WARS showed the most dramatic changes with IFNγ stimulation in protein abundance and gene expression alike (*Figure 4D*). On the other hand, the top five final candidates for M(IL-4) were LIPA, TGM2, EVL, PFKP and LRRFIP1, which showed a broad range of expression kinetics with their highest induction at different time points, implying that the related in vitro validation should be planned accordingly (*Figure 4D*). Finally, the top prioritized and the top combined-ranked proteins showed good agreement with the rankings found through the alternative correlation measures and prioritization methods discussed above, showing significant overlap (*Figure 2—figure supplements 8C–D*, *9B–C and E–F*).

To inspect more closely the molecular paths between our top candidates and drug targets, we created subnetworks connecting the top five candidates to all CVD drug targets (*Figure 4E and F*). STAT1 and TGM2 tended to connect to CVD drug targets through more established links in the literature-curated PPI network, whereas GBP1, WARS and NAMPT were mostly connected to drug targets by co-abundance edges (*Figure 4E*). Similarly for M(IL-4), we noted that TGM2, PFKP and LRRFIP1 mostly utilize PPI network edges to link to drug targets while EVL and LIPA exploit the information provided by the co-abundance links (*Figure 4F*). Mapping the shortest paths between the known CVD drug targets and top-ranked candidates thus presents us with an unbiased network-based means to select putative targets that preferentially leverage the novel information provided by the context-specific co-abundance links.

## Validation experiments identify GBP1 and WARS as potential regulators of pro-inflammatory signaling

Although a systems approach facilitates target discovery, increased or decreased genes or proteins identified by unbiased omics screening may not necessarily play causal roles. While the expression levels of certain proteins increase during the activation of human macrophages, as gauged by induction of pro-inflammatory molecules, these molecules may not contribute to this phenotypic switch. Thus, to provide mechanistic insights about these induced molecules, as well as to validate our systems approach, we performed in vitro loss-of-function experiments. Based on the shortest path mapping between top candidates and CVD targets (*Figure 4E–F*), we chose GBP1 Guanylate Binding Protein (1) and WARS (Tryptophanyl-tRNA Synthetase) as candidates for novel regulators of pro-inflammatory macrophage activation and proceeded to in vitro validation experiments. Evidence had linked WARS with vascular angiogenesis and homeostasis (*Ewalt and Schimmel, 2002*; *Otani et al., 2002*; *Wakasugi et al., 2002*). Further, genome-wide linkage studies have previously implicated this molecule in myocardial infarction (*Broeckel et al., 2002*), although subsequent genetic-epidemiological studies did not find significant associations between WARS and the risk of myocardial infarction (*Zee et al., 2005*). A more recent study that used human primary peripheral blood mononuclear cells (PBMCs) demonstrated that secretion of the full-length form of WARS is induced by pro-inflammatory stimuli, including bacterial-derived lipopolysaccharides (LPS), suggesting a role for WARS in the defense system against infection (*Ahn et al., 2017*). Here, we investigated the potential for WARS to regulate prototypical pro-inflammatory signaling proteins and cytokines in response to IFNγ. We performed small interfering RNA(si-RNA)-mediated WARS loss-of-function studies on THP-1-differentiated macrophage-like cells and human primary macrophages derived from PBMC. In THP-1 cells stimulated by IFNγ, silencing of WARS enhanced the mRNA expression of the chemokine CCL2, and the secretion of its protein (*Figure 5A*), however no such effects were observed for the cytokine TNFα (*Figure 5A*). The enhanced secretion of CCL2 in response to WARS silencing was confirmed in three different PBMC donors, where the increase in CCL2 secretion was significant at 12 hr of IFNγ stimulation (*Figure 5—figure supplement 1A*). Investigating the effect of WARS on the JAK-STAT pathway, the transcriptional levels of STAT1 and JAK2 did not change with WARS silencing (*Figure 5A*). Silencing of WARS, however, increased the phosphorylation of STAT1

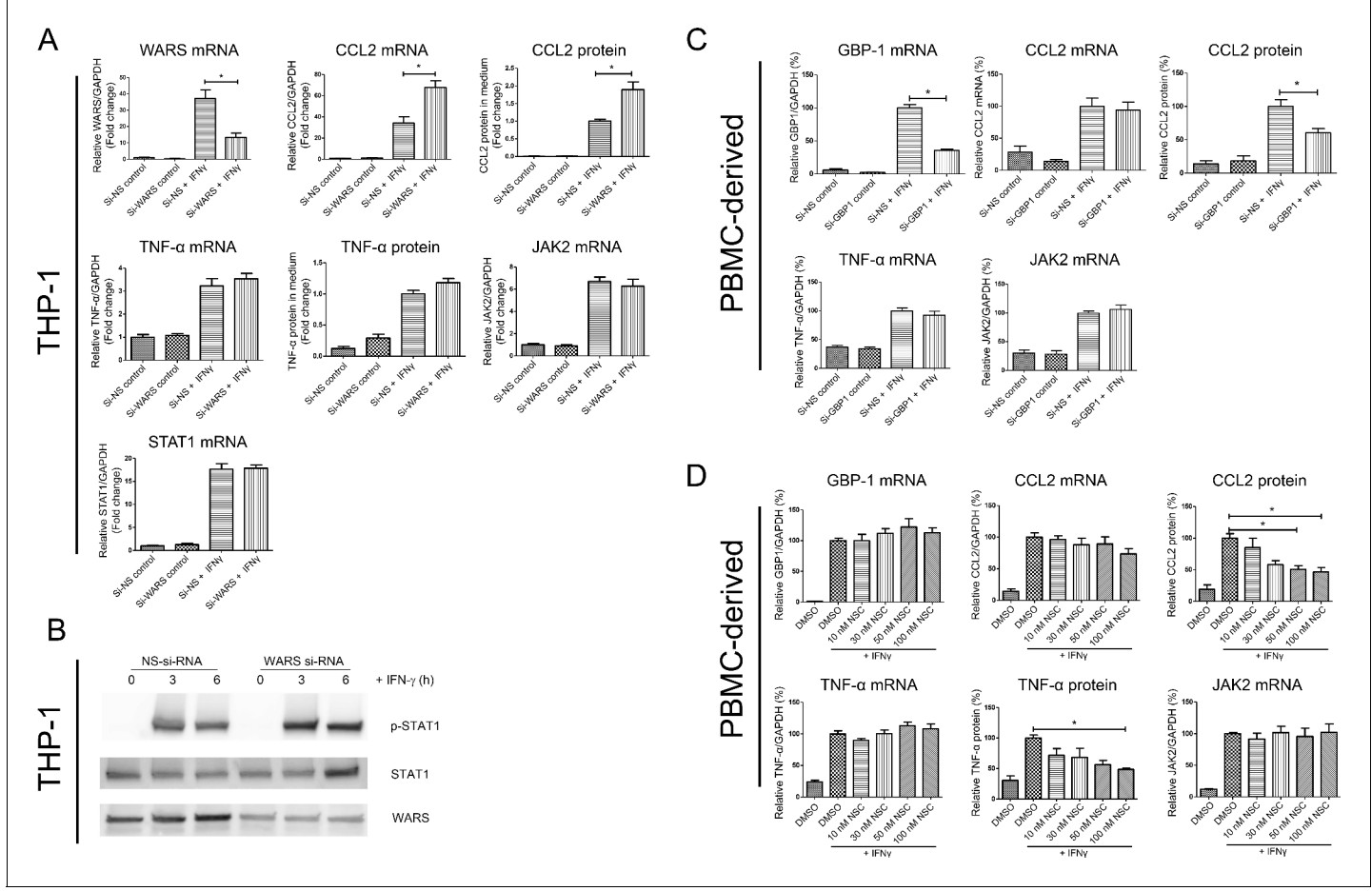

**Figure 5.** In vitro loss-of-function experiments for GBP1 and WARS. (**A**) Relative mRNA and protein expression levels for CCL2, TNFα, JAK2 and STAT1 and with WARS silencing under control and IFNγ stimulation conditions, performed on THP-1-differentiated macrophage-like cells.(n = 8–12 samples from three to four experiments) (**B**) Western blot showing protein expression levels of pSTAT1 with WARS silencing under control and IFNγ stimulation conditions, performed on THP-1-differentiated macrophage-like cells. Data shown as representative image from three experiments (**C**) Relative mRNA and protein expression levels of CCL2, TNFα and JAK2 with GBP1 silencing under control and IFNγ stimulation conditions, performed on human primary PBMC-derived macrophages (n = 9–12 samples from four human donors). (**D**) Relative mRNA and protein expression levels for CCL2, TNFα, and JAK2 under treatment with NSC756093, a GBP1-PIM1 inhibitor, in human PBMC-derived macrophages (n = 7–10 samples from four human donors) stimulated with IFNγ for 24 hr. mRNA expression levels were measured by RT-PCR and normalized by GAPDH expression. Protein in culture media was detected by ELISA. In all figures, *p<0.05, one-way ANOVA unless otherwise noted.

The online version of this article includes the following figure supplement(s) for figure 5:

**Figure supplement 1.**
**Figure supplement 2.**
**Figure supplement 3.**

at Ser701 in both THP-1-differentiated macrophage-like cells and human primary macrophages (**Figure 5B** and **Figure 5—figure supplement 1B**).

GBP1 is a member of the IFNγ-inducible GBP family, whose members are protective against bacterial (**Kim et al., 2011**) and parasite (**Selleck et al., 2013**) infection. Following a Western-type diet GBP3 and GBP6 expression levels increase during foam cell formation in mice, indirectly suggesting their role in the acceleration of atherosclerosis by hypercholesterolemia (**Goo et al., 2016**). We performed loss-of-function studies to determine the possible mechanisms through which GBP1 influences pro-inflammatory molecules. In human PBMC-derived macrophages from four different donors stimulated with IFNγ, GBP1 silencing resulted in a significant decrease in CCL2 secretion without change to its mRNA at 24 hr of IFNγ treatment (**Figure 5C**). The expression of TNFα and JAK2 did not change (**Figure 5C**). We found similar results in THP-1 cells (**Figure 5—figure supplement 1C**).

To assess the effect of currently available therapeutic inhibitors targeting GBP1, we used NSC756093 that interrupts the interaction between GBP1 and proto-oncogene serine/threonine-protein kinase (PIM1), the inhibition of which could potentially revert paclitaxel resistance in cancer cells (*Andreoli et al., 2014*). Human primary macrophages were pretreated with NSCT56093 (10 to 100 nM) and then stimulated with IFNγ for 24 hr. We observed a dose-dependent decrease in the secretion of both CCL2 and TNFα (*Figure 5D*), with no change to their mRNA levels and to the mRNA levels of JAK2 and GBP1 itself (*Figure 5D*). Silencing of GBP1 decreased the expression of JAK2 and phospho-STAT1 at 6 hr of IFNγ stimulation in THP-1 macrophage-like cells (*Figure 5—figure supplement 1D*).

With the hypothesis that molecules that arise from in silico predictions seeded with known CVD targets and found to play specific regulatory roles in pro-inflammatory macrophage activation are likely to be CVD therapeutic targets, we next sought to investigate their structural 'druggability' in silico. The top five candidates of pro-inflammatory macrophage activation all contained binding sites suitable for small molecules, with at least 50% tractable structures and 20% druggable structures (see Materials and methods, *Figure 5—figure supplement 2A–B*). Among these, WARS and GBP1 were the two candidates with the highest percentage of druggable structures (*Figure 5—figure supplement 2B*).

## Discussion

The search for new molecular connections between inflammation and CVD is continuing at an ever-increasing pace. Network medicine approaches utilizing large-scale interactomes hold the key to the efficient identification of novel therapeutic solutions targeting pro-inflammatory macrophage activation in CVD. Currently available interactomes, however, are hindered in their usefulness since they are by design the aggregation of all possible interactions from diverse cell states, which obscures any tissue, cell type, experiment or disease-specific information. One approach to address this lack of specificity and incompleteness is to enhance the interactome with context-specific information from proteomics measurements. To capitalize on the potential for interactome networks to expedite CVD drug target discovery research, we combined our protein co-abundance networks derived from a cell-specific, macrophage activation in vitro model with the 'all-purpose', generic PPI network. We interrogated the resulting combined interactome to predict and highlight new regulators of macrophage activation and potential targets of CVD as well as studied the potential the mechanisms by which they might regulate pro-inflammatory biomarkers, and experimentally validated some of these mechanisms.

When proposing new sources for protein interactions, two questions must be asked: (i) Does the new network add novel information to the existing one?; and (ii) Is this novel information biologically relevant? The first question is addressed by the 'fill-in-the-blanks' effect of the co-abundance edges, where the sparse portions of the PPI network were filled in by co-abundance networks, potentially addressing the incompleteness and investigation bias of the PPI network. Previously, co-expression networks were suggested to represent a complementary tool to the PPI network (*Vella et al., 2017*), and protein abundances obtained through mass spectrometry and direct protein contacts detected by crosslinking and mass-spectrometry were found to be complementary (*Solis-Mezarino and Herzog, 2017*). Interestingly, each stimulation condition was distinct in regards to which part of the PPI network they filled in. We also noted that the M(IL-4) co-abundance network had more overlap with M(-). Addressing the second question, our assumption is that proteins that are co-regulated are more likely to be involved in similar pathways (*Kustatscher et al., 2017*). Indeed, the co-abundance edges were significantly enriched in interactions based on binary interactomes, co-complex associations and shared functional annotations. While direct binary interactions were less strongly represented among co-abundance edges, the indirect co-associations evidenced by the enrichment of multiple co-complex resources suggests that the real value of co-abundance edges might lie in their use as co-complex resources and not necessarily direct physical interactions.

It has been proposed that network proximity is a good proxy of therapeutic effect (*Guney et al., 2016*). Indeed, we found that network distance-based measures fared better with the addition of context-specific co-abundance edges, and that co-abundance edges rendered the PPI network more navigable for the drug targets. The prediction performance of novel drug targets based on network proximity measures showed a significant improvement with the addition of co-abundance edges.

The prediction performance further increased when co-abundance edges were given more prominence, which suggests that the information provided by the co-abundance network is beneficial and adds positive value to the PPI network when predicting drug targets. Remarkably, the prioritized candidates also captured disease biology, especially inflammation, innate immune response and CAD signatures. This suggests that the drug target candidates themselves may play a direct role in CVD or be involved in the inflammatory mechanisms leading to it. We note that, while co-abundance edges were built systematically, filtered with statistical rigor, and as a whole contributed to the increase in drug target prediction accuracy, they were not experimentally validated in this study and therefore are not deemed high-confidence edges individually.

Aside from hallmark pro-inflammatory pathways such as MAPK and NF-κB, the top ranked candidates for M(IFNγ) were enriched in various pathways associated with atherosclerosis and other vascular disorders, including Wnt, FGF, adrenergic, IGF, Akt, and ras. IFNγ was suggested to have direct effects on cardiomyocytes through β-adrenergic signaling (*Levick and Goldspink, 2014*), PI3K-Akt pathway has been posed as a key regulator in macrophage metabolism (*Brenner et al., 2016*). The enriched pathways of the top ranked candidates for M(IL-4) showed a breadth of evidence for the alternatively activated, pro-resolving macrophage polarization. Among these, Rho GTPase activation has been associated with M2-like anti-inflammatory macrophage phenotype (*Aflaki et al., 2011*). SUMOylation was linked to anti-inflammatory signals and suggested as a potential target pathway for the modulation of inflammation (*Leitinger and Schulman, 2013*; *Tugal et al., 2013*). Elevated levels of Fas (CD95) expression in anti-inflammatory macrophages was associated with proangiogenesis in the eye (*Zhao et al., 2013*). Alternatively activated macrophages were shown to activate Wnt signaling pathway and increase β-catenin expression in epithelial cells (*Cosín-Roger et al., 2013*). The expression of several scavenger receptors was increased in alternatively activated macrophages (*Canton et al., 2013*). This pathway-based evidence suggests that co-abundance networks indeed influence the PPI network and the resulting drug target prioritization reflects the biology of the respective macrophage activation.

The correlation between gene and protein expression may be low in multi-cellular organisms (*Kustatscher et al., 2017*; *de Sousa Abreu et al., 2009*), which we also observed in our datasets (*Figure 5—figure supplement 3A*). To ameliorate this issue, we used a combined ranking of rankings to effectively normalize the ranking distributions and address the possible delay between gene expression and protein expression (*Liu et al., 2016*). The resulting filtered list of candidates indeed showed the desired closeness to known CVD drug targets. When measuring the network distance between the final list of candidates and CVD targets, we used two measures that offer a complementary view of network proximity: Average shortest distance is a useful summary metric that quantifies the distance between candidates and targets while shortest distance distributions provide a more detailed view enabling us to see which number of links between candidates and targets tend to be over-represented compared to random. While minimum (or closest) shortest distance has been found to outperform other proximity measures (*Guney et al., 2016*), we chose to use average shortest distance as the metric of choice in our drug target prioritization method since minimum shortest distances are confined to integer numbers resulting in redundancy of ranks, which convolutes the overall rankings for prioritization. Considering potential limitations of shortest path based prioritization methods such as ignoring biologically meaningful alternative longer-range paths between candidates and drug targets, we also ran our workflow on global dynamical prioritization measures, which displayed similar results, suggesting that the improvement of target prediction with context-specific information from co-abundance networks is independent of the prioritization metric used.

To validate our systems approach to the discovery of new regulators of macrophage activation, we chose GBP1 and WARS from the subnetworks of top five candidates and CVD drug targets. In particular, GBP1 and WARS stood out as the candidates that were linked to the majority of drug targets by novel co-abundance links, hence potentially having many undiscovered downstream mechanisms related to CVD and inflammation. It is important to note the opposite influences of GBP1 and WARS on JAK-STAT signaling, particularly on the pro-inflammatory cytokine CCL2 and pSTAT1. Our siRNA silencing and inhibitor treatment experiments suggest that WARS has a protective effect, suppressing these downstream pro-inflammatory markers, whereas GBP1 has an exacerbating effect on inflammation, enhancing their expression, along with another pro-inflammatory molecule, JAK2. Overall, the downstream effect of GBP1 points at a possible feedback regulation on JAK2-STAT1-

CCL2 signaling (*Figure 5—figure supplement 3B*). Other GBP family members, GBP2, GBP3, GBP4 and GBP6, were also found in our proteomics measurements, showing higher induction after 24 hr, with GBP2 displaying similar expression patterns to GBP1, while GBP3 and GBP6, the two family members found to be induced in macrophage foam cells (*Goo et al., 2016*), have a distinct pattern characterized by a sharp drop in protein expression at 12 hr and then a sharp increase at 24 hr (*Figure 5—figure supplement 3C*). Coupled with evidence from our network-based prioritization, this suggests that other members of the GBP family with similar expression patterns to GBP3 and GBP6, including the top-prioritized GBP1, might also play a role in atherosclerosis mediated by hypercholesterolemia.

The present work can be improved in multiple directions. While we adopted a simple weighted integration method of the PPI network and co-abundance networks to take into account the predictive potential of each network, similar to the weighted network integration method demonstrated in the study by *Valentini et al., 2014*, many alternative filtering strategies remain. As an alternative to combining the PPI network and co-abundance networks, we explored a filtering strategy where we removed the negative correlation edges in addition to adding the positive correlation ones. However, the number of edges that were already present in the PPI was limited to a few hundreds 114, 271 and 105 edges for M(-), M(IFNγ) and M(IL-4), respectively), therefore the removal of these did not result in a substantial increase in the ROCs, with limited improvement for M(IFNγ) and M(IL-4) in terms of AUROCs (86.77% to 86.78% for M(IFNγ) and 85.18% to 85.20% for M(IL-4). In other words, the improvement caused by the removal of a few hundred edges was mostly overshadowed by the addition of many more positive correlation edges. This suggests that the main shortcoming of the PPI is incompleteness, which is thought to be a large percentage of all possible edges, rather than noise and false positives, which make up a much smaller portion. Another constraint regarding the building of co-abundance networks is that we used only the common proteome between M(-), M(IFNγ) and M(IL-4) and applied a PSM cutoff of >10, which reduces the size of the co-abundance networks. While this ensures that we work with the strongest signals, relaxing these criteria might result in the discovery of additional key targets.

In summary, the present study demonstrates that proteomics-derived co-abundance edges introduce context specificity to the PPI network and significantly improve the prediction of drug targets related to the biology in question. Indeed, co-abundance networks derived from IFNγ- and IL-4-stimulated macrophage-like cells resulted in the network proximity-based prediction of GBP1 and WARS as potential regulators of pro-inflammatory macrophage activation. Our in vitro loss-of-function studies involving human primary macrophages verified the role of these targets in pro-inflammatory signaling as regulators of CCL2 and JAK2 as well as STAT1 phosphorylation. Overall, our workflow has general applicability and can serve as the blueprint for subsequent studies to combine proteomics data with context-independent interactomes to extract cell type- and experimental condition-specific information for the purpose of identifying targetable pathways and molecules in the context of complex pathobiologies such as CVD.

# Materials and methods

## Construction of the literature-based protein-protein interaction (PPI) network

To represent the current knowledge on human protein-protein interactions as a literature-based PPI network, we compiled a comprehensive list of PPIs with experimental evidence from various databases (*Menche et al., 2015*) including the following types of interactions: (i) regulatory interactions (*Matys et al., 2006*), (ii) high-quality binary PPIs tested via high-throughput yeast two-hybrid (Y2H) systems, obtained from multiple publications (*Rolland et al., 2014*; *Rual et al., 2005*; *Stelzl et al., 2005*; *Venkatesan et al., 2009*; *Yu et al., 2011*) and public databases (*Aranda et al., 2010*; *Ceol et al., 2010*), (iii) literature-curated PPIs identified by affinity purification followed by mass spectrometry (AP-MS), Y2H, low-throughput experiments, and protein three-dimensional structures (*Aranda et al., 2010*; *Ceol et al., 2010*; *Keshava Prasad et al., 2009*; *Stark et al., 2011*; *Zhang et al., 2013*), (iv) metabolic enzyme-coupled interactions (*Lee et al., 2008*), (v) protein complexes derived from a variety of experimental tools, from co-immunoprecipitation to co-sedimentation and ion exchange chromatography (*Ruepp et al., 2010*), (vi) kinase-substrate interactions

derived from high-throughput and literature-curated low-throughput experiments (*Hornbeck et al., 2012*), and (vii) signaling interactions from both high-throughput and literature curation (*Vinayagam et al., 2011*). After the removal of duplicated edges, the resulting PPI network contained 170,303 interactions between 14,213 proteins. For our analyses, we discarded the isolated nodes that only self-interact and used the largest connected component (LCC) of this network, which has 170,253 interactions and 14,115 proteins. The resulting network has an average degree <k > of 24.12, an average clustering coefficient <C> of 0.210, a diameter of 12 and an average shortest path length of 3.54 (*Supplementary file 1*). To address the investigation bias and eliminate noise due to indirect associations from protein complexes wherever possible within the PPI network, we systematically removed the edges from individual, low-throughput experiments from literature and co-complex interactions. The removal of low-throughput edges resulted in 13,604 proteins and 147,295 interactions and the subsequent removal of co-complex edges resulted in 13,568 proteins and 125,495 interactions. As it yields the best performance (*Figure 2—figure supplement 3*), this 'de-noised' PPI with the low-throughput and co-complex interactions removed was used throughout the manuscript.

## TMT sample preparation and liquid chromatography tandem mass spectrometry

We have detailed the IFNγ and IL-4 stimulation conditions, cell culture experiments and six-plex tandem mass tagging (TMT) sample preparation methods previously in *Iwata et al., 2016*. The TMT peptide samples were analyzed using high-resolution and accuracy LTQ-Orbitrap Elite (Thermo Scientific) and subsequently annotated using the SEQUEST search algorithm via the Proteome Discoverer (PD) Package (version 1.3, Thermo Scientific) (*Eng et al., 1994*) as described previously (*Iwata et al., 2016*). Master proteins with two or more unique peptides were used for TMT reporter ratio quantification. For each peptide-spectrum match (PSM), the TMT ion channel intensities were normalized to the time-zero channel. Protein abundances were then calculated by taking the median of their corresponding PSM ratios (*Dayon et al., 2008*).

## Construction of co-abundance networks

To ensure that the subsequent co-abundance networks are built out of the proteins detected with high confidence, we further filtered the list of proteins to those with more than 10 PSMs. For each of these proteins with PSM >10 (2555, 2586 and 2695 proteins for M(-),M(IFNγ) and M(IL-4), respectively), we then extracted the abundance profile, which consists of six time points (0, 8, 12, 24, 48 and 72 hr). Then, for every pair of proteins, we calculated the Pearson correlation coefficient ($r$) between their abundance profiles. This resulted in a weighted, complete graph where the edge weights were given by Pearson's $r$ values. As a robust way of comparing these correlation values against a null model representing the expectation by chance, we created randomized datasets whereby the abundance profile vectors were shuffled for a large number of realizations, and calculated permutation-based empirical p-values. In this case, the empirical p-value (P*) was calculated as $P* = r_>/N$ where N = 300 is the total number of permutations performed and $r_>$ is the number of permutations where the permuted $r$ was higher than the real $r$. In other words, P* is the probability of encountering a higher value of $r$ in the permuted data than the observed $r$. We then adjusted the empirical p-values for multiple-testing correction using the Benjamini-Hochberg (BH) procedure to control for the false discovery rate (FDR). This resulted in a Pearson correlation and adjusted empirical p-value pair (r, Q*) for all possible edges. Finally, to filter the network to preserve only the most high-confidence co-abundance edges, we set an edge weight threshold. For the selection of this threshold, we performed a sensitivity analysis where we plotted the density of the resulting co-abundance network as a function of the adjusted empirical p-value. At an FDR of 1%, we selected the highest Pearson correlation that maintained the network density, that is 0.90, as the edge weight cutoff (*Figure 1—figure supplement 1A*). As additional correlation measures, we calculated biweight midcorrelation and partial correlation using the WGCNA package (*Langfelder and Horvath, 2012*) and ppcor package (*Kim, 2015*), respectively, in R. Baseline (M(-)) abundances were controlled for in the partial correlation calculation. Resulting two-sided p-values were adjusted for multiple testing using the BH procedure, and a correlation threshold of 0.90 (p<0.01) was chosen for consistency.

## Measuring the biological relevance of co-abundance links

We tested the biological relevance of the co-abundance networks by quantifying their edge overlap with the external protein-protein interaction networks below:

(a)Binary interactions: We obtained human binary interactions from HINT database (http://hint.yulab.org/; *Das and Yu, 2012*), which had 55,493 interactions (retrieved March 2017), as well as two interactomes based on Y2H assays: HI-II-14 (*Rolland et al., 2014*) and HI-III-16 (Unpublished, from the Center for Cancer Systems Biology), which contain 11,603 and 48,229 interactions, respectively, after mapping from UniProt IDs to gene symbols. (b) Affinity Purification-Mass Spectrometry-based interactions: We used two recent sets of interactomes based on AP-MS measurements. BioPlex 2.0 (http://bioplex.hms.harvard.edu/; *Huttlin et al., 2015*) contained 56,553 interactions and (*Hein et al., 2015*) contained 27,380 interactions overall. (c) Co-complex interactions were obtained from HINT (*Das and Yu, 2012*) co-complex interactome consisting of 121,546 interactions (retrieved March 2017) and a curated co-complex association network (*Woodsmith and Stelzl, 2014*) consisting of 74,131 interactions. We also derived interactions based on co-complex membership using the CORUM (http://mips.helmholtz-muenchen.de/corum/; *Ruepp et al., 2010*) protein complex database, which resulted in 47,378 interactions (retrieved July 2017). Lastly, we used hu.Map, a recent large-scale protein complex map resulting from the integration of over 9000 MS experiments, which contained 35,375 interactions after setting an interaction probability threshold of 0.265, which is deemed high-confidence (*Drew et al., 2017*). (d) We constructed GO term networks for the three branches of the GO, namely the Biological Process (GO:BP), Molecular Function (GO:MF) and Cellular Component (GO:CC) by taking the one-mode projection of the gene-term bipartite network on the gene component, that is by connecting genes based on their shared GO terms. We excluded the inference-based evidence types IPI (inferred from physical interaction) and IEA (inferred from electronic analysis), and ND (not described). To ensure that the GO networks adhere to relatively specific GO terms, we considered terms with less than 100 associated genes. This resulted in 11,361 nodes and 631,799 edges for GO:BP, 7586 nodes and 162,154 edges for GO:MF, and 6044 nodes and 196,936 edges for GO:CC, after mapping Entrez IDs to gene symbols. The GO database was downloaded from http://www.geneontology.org/ in August 2015.

To calculate the edge overlap, we pruned each external interaction network to contain only the proteins present at the intersection of that network and the co-abundance network. We used two-sided Fisher Exact test to calculate the enrichment odds ratios, confidence intervals and p-values.

## Curation of CVD drug targets

We retrieved the known drug targets for cardiovascular diseases by querying iCTNet2 (*Wang et al., 2015*) (http://apps.cytoscape.org/apps/ictnet2; retrieved November 2015) for phenotype-gene, gene-drug and disease-drug interactions. To cover a broad range of cardiovascular diseases, we selected the 'cardiovascular system disease' phenotype (ID: 1287) and all its subcategories and included the results from both the GWAS catalog and the Online Mendelian Inheritance in Man (OMIM) database. For the drug associations, we queried iCTNet2 for disease-drug and gene-drug interactions from DrugBank (*Wishart et al., 2006*) and the Comparative Toxicogenomics Database (CTD) (*Mattingly et al., 2003*). This resulted in a tripartite network of cardiovascular system diseases and their related genes and targets. We excluded isolated cardiovascular system diseases with no drug or gene interactions, or only one type of interaction. We then eliminated diseases and extracted the largest connected component of the remaining network to arrive at the final bipartite network of 268 drugs and 283 drug targets (*Figure 1—figure supplement 1C*). Of the 283 drug targets 251 were mapped onto the PPI network and 52 were represented in the co-abundance networks. While the iCTNet database also has protein-protein interactions, they were not used in our study to avoid circularity.

## Drug target prioritization and its performance assessment

We used a drug target prioritization method based on the topological proximity of candidate genes to all seed genes in the molecular interaction network, where seed genes are defined as the known cardiovascular drug targets. In particular, the proximity score is inversely proportional to the network distance between candidate genes and seed genes, where the proximity score *PS(c)* for each candidate *c* is defined as

$$PS(c) = \sum_{s \in S} \frac{I(s)}{d_{cs} + 1}$$

where $I(s)$ is the relative weight, or importance, of seed $s$, $d_{cs}$ is the shortest network distance between candidate gene $c$ and seed gene $s$ and $S$ is the set of all seed genes. The non-Euclidean network distance $d_{cs}$ is measured in terms of the number of links. As the drug target data lacks information about their relative importance, we weighted the seed genes equally, assigning them the relative weight $I(s)$ value of 1.0. Using the proximity score, we ranked all the proteins in the PPI network. As a measure of the prediction performance, that is the rate of capturing true positives, of our prioritization method, we plotted the receiver operating characteristic (ROC) curves and calculated the area under the ROCs (AUROC). We implemented k-fold (k = 7 was chosen for a reasonable test set size, unless otherwise noted) cross-validation to determine the statistical significance of the difference between AUROC values, and as the datasets were partitioned into folds consistently across PPI and PPI + CoA, a paired t-test was used to compare the two cases and two-tailed p-values were reported. For the addition of random edges used as negative control in the prediction performance assessment, we adopted a degree-preserving randomization strategy. As there is typically a small number of very highly connected proteins in the PPI network, to avoid repeatedly selecting the same proteins in lieu of these highly connected nodes, all proteins were logarithmically binned according to their degree. Edges were then established between pairs of proteins uniformly randomly picked from their respective degree bin representing a pool of similar degree proteins. In addition to ROC curves, we additionally plotted precision-recall curves as they can provide additional information for imbalanced datasets where positives are rare. As additional prioritization measures, we implemented RWR and DADA as described in *Erten et al., 2011*, and Kernel distance *dk(c)* per candidate $c$ was defined as

$$d_k(c) = -ln \sum_{s \in S} e^{-\frac{(d_{cs}+1)}{|S|}},$$

where $d_{cs}$ is the shortest network distance between candidate gene $c$ and seed gene $s$ and $S$ is the set of all seed genes.

## Optimization of the prioritization method

In order to further optimize the prediction performance, we sought to distinguish between co-abundance and PPI links. For this, we redefined network distance $d_{cs}$ to accommodate the link weights such that co-abundance link weights $w_{CoA}$ and PPI link weights $w_{PPI}$ have different values, which effectively results in two different types of links. The new network distance $d^*_{cs}$ between a candidate protein $c$ and a seed protein $s$ therefore becomes

$$d^*_{cs} = \sum_{C \in E_{CoA}} w_C n_C + \sum_{P \in E_{PPI}} w_P n_P,$$

where C (P) denotes co-abundance (PPI) links, $E_{CoA}$ ($E_{PPI}$) is the set of all co-abundance (PPI) links, $w_C$ ($w_P$) is the weight of co-abundance (PPI) links, and $n_C$ ($n_P$) is the number of co-abundance (PPI) links in the shortest path between $c$ and $s$. Since the proximity score $D(c)$ is inversely proportional to the distance between candidates and seeds, the ratio between the weights of the two types of links modulates the relative importance of these link types in the prioritization scheme. Under the regime $w_C/w_P < 1$, the co-abundance network is given the advantage, influencing the enriched network topology more than PPI whereas under for $w_C/w_P > 1$, PPI has more influence than the co-abundance component. In the case of $w_C/w_P = 1$, $d^*_{cs}$ simplifies to $d_{cs}$, corresponding to the case where co-abundance and PPI links are treated equally. With the weight ratio as our single parameter, we performed a scan where we calculated the area under the ROC curve (AUROC) as a function of this ratio in the range [0, 10], which displayed a maximum in this range (*Figure 2—figure supplement 4A*) indicating that this procedure of weighing the two types of links is amenable to optimization. We also implemented k-fold (k = 7 was chosen for a reasonable test set size, unless otherwise noted) cross-validation in the optimization procedure, and selected the weight ratio that yields the highest mean AUROC across all folds for each condition, to be used for subsequent analyses throughout the

manuscript. These values were found to be 0.4, 0.4 and 0.1 for M(-), M(IFNγ) and M(IL-4), respectively.

## In silico validation datasets

For the external validation of drug targets, we downloaded drug-target interactions and drug indications from the DrugCentral database (*Ursu et al., 2017*) (http://drugcentral.org/; datasets were time stamped 04/25/2017). After removing drugs without associated gene symbols, we obtained 2272 targets (1444 of which were mapped to the PPI and 324 of which were in the co-abundance networks) for the 'DrugCentral' dataset representing all drug-target interactions in the DrugCentral database and 599 targets (229 of which were mapped to the PPI and 23 of which were in the co-abundance networks) for the 'T_clin' dataset representing the targets with known mechanisms of action. To acquire the CVD related targets from DrugCentral, we mined the drug indications data for the keywords 'arterio', 'athero', 'artery', 'cardi', 'coronary' and 'heart'. This resulted in 502 CVD targets (330 of which were mapped to the PPI and 23 of which were in the co-abundance networks) for DrugCentral, 115 CVD targets (55 of which were mapped to the PPI and 1 of which was in the co-abundance networks) for the 'T_clin' dataset. In addition to the drug-target interaction and drug indication data from DrugCentral database, we downloaded detailed target information from TCRD database (*Nguyen et al., 2017*) (http://juniper.health.unm.edu/tcrd/; Version 4.6.2), which contains information about the top five associated diseases and targets are categorized into four categories based on their development and druggability levels. We considered the targets belonging to the 'Clinical' (approved drugs) and 'Chemical' (drugs that satisfy the activity thresholds outlined in the TCRD website - http://juniper.health.unm.edu/tcrd/) categories. Mining the top 5 diseases of TCRD Clinical and Chemical datasets for the CVD related keywords above, we obtained 53 CVD targets (52 of which were mapped to the PPI and 9 of which was in the co-abundance networks) for Clinical and 121 CVD targets (116 of which were mapped to the PPI and 27 of which was in the co-abundance networks) for the Chemical category.

We used three datasets to test the relevance of the prioritized proteins to inflammatory processes, innate immune response and cardiovascular disease. For inflammation, we looked for the enrichment of the top-ranked proteins in the inflammatome signature (*Wang et al., 2012*), which includes common rodent inflammatory signatures from 12 expression profiling datasets corresponding to nine different tissues and 11 disease models. Overall, this dataset contained 2483 genes that comply with a consensus p-value threshold. Of these 1942 were mapped to the PPI network and 555 of which were in the co-abundance networks. The innate response genes were obtained from the InnateDB database (http://www.innatedb.com/; *Lynn et al., 2008*), which is a knowledge base for the mammalian innate immune response captured through contextual manual curation. It includes over 18,000 interactions between over 1500 proteins (retrieved in April 2016), 872 of which were mapped to the PPI network and 233 of which were in the co-abundance networks. For cardiovascular disease signature, we used the CADGene database (http://bioinfo.life.hust.edu.cn/CADgene/; *Liu et al., 2011*), which includes genes implicated by genome wide association studies (GWAS) as well genes obtained by manual literature curation. Overall, it included 604 genes related to coronary artery disease, 574 of which were mapped to the PPI network and 97 of which were in the co-abundance networks. Finally, to further expand our list of CAD genes, we used the 'CAD1000G Extend' dataset described in *Zhao et al., 2016*, containing 881 CAD genes (560 of which were mapped to the PPI network and 148 of which were in the co-abundance networks) based on the GWAS catalog, additional candidate genes identified through the CARDIoGRAM-C4D study based on Metabochip data and therefore not included in the GWAS catalog, further supplemented by genes from the 1,000 Genomes study (*1000 Genomes Project Consortium et al., 2015*).

## Pathway enrichment and construction of pathway networks

For the pathway enrichment analysis, we used ConsensusPathDB (*Kamburov et al., 2011*) (pathway data retrieved from http://consensuspathdb.org/ in February 2017) and considered the canonical pathways from KEGG, Biocarta, and Reactome (*Chowdhury and Sarkar, 2015*). This resulted in 301 pathways from KEGG comprising 7121 genes, 252 pathways from Biocarta comprising 1408 genes, and 1764 pathways from Reactome comprising 10,095 genes, which together defined a pathway space of 2317 pathways consisting of 11,447 genes. We tested the top prioritized proteins for

pathway enrichment by a hypergeometric test and adjusted for multiple comparisons using the Benjamini-Hochberg method for controlling false discovery rate (FDR). The cutoff to determine the top prioritized proteins was based on the permutation of p-values whereby an empirical p-value was calculated for each rank for N = 10,000 random realizations. Pathways with FDR adjusted p-value (q-value) <0.05 were considered significantly enriched. The pathway networks represent pathways as the nodes and the shared genes between pathways as the edges. Node size corresponds to –log (q-value) and edge weight (thickness) corresponds to the gene overlap between pairs of pathways measured by the Jaccard index $J$, which is defined as

$$J = \frac{s_A \cap s_B}{s_A \cup s_B}$$

where $s_A$ and $s_B$ are the sets of top prioritized proteins that belong to pathway $A$ and pathway $B$, respectively. We calculated $J$ for all enriched pathway pairs and discarded edges with $J$ values less than 0.1 in the visualization for clarity. The network visualizations were made using Gephi v0.8.2 (*Bastian and Heymann, 2009*).

## Combined ranking and determination of the final set of candidates

To condense the list of network-prioritized candidates, which consists of all of the proteins in the interactome, further into a smaller list supported by proteomics and transcriptomics evidence, we followed a three-layer filtering procedure. The combined ranking was achieved as the result of (i) the rank according to the network prioritization, (ii) the rank according to the relative protein abundance difference with respect to baseline M(-) over all time points, and (iii) the gene expression fold change with respect to baseline M(-) from human macrophage transcriptome data (*Xue et al., 2014*). We first determined the optimal threshold on the network-based prioritization ranking that would maximize the sum of the sensitivity (true positive rate) and specificity (1 – false positive rate). Based on the ROC curves obtained after optimization of weight ratios, we found these optimal threshold values to be the top 2971 candidates for M(IFNγ) and the top 3496 candidates for M(IL-4) (*Figure 4—figure supplement 1A–B*). Second, we filtered these top-N candidates with the top 500 candidates with the highest relative abundance in the proteomics data for each time point, with respect to the baseline M(-). As the third step of the filtering procedure, we use the top 500 genes with the highest fold change for IFNγ and IL-4 stimulated macrophages from an extensive dataset of human macrophage activation transcriptomes (*Xue et al., 2014*). These three steps resulted in 43 and 49 candidates for M(IFNγ) and M(IL-4), respectively. Finally, we re-ranked this final list of candidates using a combination score based on the network-based prioritization rank, relative abundance rank (defined as the total difference between the relative abundance profiles between IFNγ or IL-4 stimulated and baseline macrophages), and expression fold change rank, such that the final combined rank is given by

$$\Re(comb.) = \Re((\Re(prior.) + \Re(abun.) + \Re(expr.))/3).$$

We confirmed that the final combined ranking is robust with respect to the choice of top-N ranked expression and abundance (*Figure 4—figure supplement 1C*). The final combined ranking was used in candidate selection for in vitro silencing experiments.

## Measuring the average distance between top target candidates and CVD drug targets

The network closeness of the candidate proteins filtered with the combined ranking to CVD drug targets was measured in terms of the average shortest distance. The average shortest distance D to CVD drug targets was measured by calculating the average shortest distance between each candidate protein $c$ and all drug targets $t$ and then averaging over all candidate proteins $c$ such that

$$d_c = \frac{1}{N_t}\sum_{t \in T} d_{ct}$$

and

$$D = \frac{1}{N_c} \sum_{c \in C} d_c,$$

where $d_{ct}$ is the shortest distance between c and t and $C$ and $T$ are the sets of proteins in the target candidates and CVD drug targets, respectively. To compare this average shortest distance value to what would be expected by chance, the average shortest distance of the same number of randomly selected proteins to CVD drug targets was calculated for N = 1000 realizations. To control for degree bias, the random protein selection was done in a degree-preserving manner where all proteins were binned according to their degree and random proteins were selected uniformly at random from their corresponding degree bin. Finally, z-scores and empirical p-values were calculated by

$$z = \frac{D - \langle D_r \rangle}{\sigma_{D_r}}$$

and

$$p_{emp.} = P(D_r < D),$$

respectively, where $D_r$ is the average shortest distance of a randomized instance, $\langle D_r \rangle$ is the mean of the average shortest distance of all randomized instances, and $\sigma_{D_r}$ is their standard deviation. Network measures including shortest distances and centralities were calculated using the NetworkX package (*Hagberg et al., 2008*) v1.9 in Python v2.7.10.

## In silico druggability assessment

To assess the potential of candidates of interest to be drug targets, we used the DrugEBIlity database (https://www.ebi.ac.uk/chembl/drugebility/) (version 3.0), which predicts the structural druggability of a molecule by how suitable its binding sites are for small molecules under the Lipinski's Rule of 5 requiring at most 10 hydrogen bond acceptors, at most five hydrogen bond donors, and a molecular weight 500 Da or less. 'Tractability' is a more relaxed criterion compared to druggability, requiring at most 15 hydrogen bond acceptors, at most eight hydrogen bond donors, and a molecular weight between 200 Da and 800 Da. 'Ensemble druggability' is the strictest criterion where the average of druggability score is calculated under different machine learning models.

## In vitro experimental setup

### Cell culture

#### THP-1 cells

THP-1 monocyte cell line was purchased from ATCC (Catalog TIB-202) in liquid nitrogen vapor phase frozen condition. THP-1 cells were originally generated from the peripheral blood monocytes from 1 year-old male infant with acute moncytic leukemia. STR profile of THP-1 includes Amelogenin: X,Y; CSF 1PO: 11,13; D13S317:13; D16S539:11,12; D5S818: 11,12; D7S820:10; THO1: 8, 9.3; TPOX:8,11; vWA:16. THP-1 cells were cultured with Roswell Park Memorial Institute (RPMI) 1640 medium supplemented with 10% FBS, 0.05 mM 2-mercaptoethanol, penicillin and streptomycin in cell culture incubator at 37°C (95% air and 5% CO2). THP-1 cell concentration remains at $2–8 \times 10^5$ cells per milliliter before the subculture every 2–3 days. The mycoplasma contamination testing was negative by using MycoAlert PLUS Mycoplasma Detection Kit (LONZA). THP-1 cells were plated in 12-well plates at $1.0 \times 10^6$ per well and differentiated into macrophage-like cells by stimulation with 200 ng/mL of PMA (Sigma-Aldrich P8139) for 48 hr.

#### Human peripheral blood mononuclear cells (PBMC)-derived macrophages

PBMCs were isolated from blood buffy coat (Research Blood Components, Brighton, MA) using lymphocyte separation medium (LSM, MP Biomedicals) as described previously. PBMCs ($5 \times 10^6$ cells/well) were cultured in six-well culture plates and maintained in RPMI supplemented with 5% human serum and penicillin/streptomycin at 37°C(5% $CO_2$) for 7–10 days before use. Human PBMC-derived macrophages were pretreated with GBP-1/PIM1 inhibitor NSC756093 (10–100 nM, Axon Medchem, Catalog# Axon2393) for 2 hr, then stimulated with human 10 ng/ml IFNγ (R and D systems) for 12 hr.

## siRNA silencing of WARS and GBP1

Silencer Select validated siRNA for human WARS was purchased from Thermo Fisher Scientific (Catalog# 439085). SMARTpool ON-TARGET plus Human GBP1 siRNA oligos were from GE Healthcare Dharmacon (L-005153). Si-RNA transfection on THP-1 cells or PBMC-derived macrophages was performed by using Magnetofection SilenceMag (OZBIOSCIENCES, San Diego) at final concentration of 50 nM. 48 hr after transfection, macrophages were stimulated with human 10 ng/ml IFNγ (R and D systems) for 6–24 hr before further experiments.

## RNA extraction and RT-PCR

Total RNA were extracted using an Illustra RNAspin Mini kit (GE Healthcare, Piscataway, NJ) and cDNAs were synthesized using a high capacity cDNA reverse transcription kit (Applied Biosystems, Carlsbad, CA). Real-time PCR was performed using Taqman probes for WARS, GBP-1, CCL2, TNFα, JAK2, STAT1, and GAPDH on a 7900HT fast real-time PCR system (Applied Biosystems). Relative expression of each gene was normalized by GAPDH.

## ELISA

CCL2 and TNFα proteins in culture medium from macrophages were detected by ELISA kit purchased from R and D systems (Minneapolis, MN).

## Western blot

Macrophages whole cell lysate were prepared using RIPA buffer containing protease inhibitor (Roche). Total protein was separated by 4–20% Mini-PROTEAN TGX Precast Gel and transferred using the iBlot Western blotting system (Life Technologies). Primary antibodies against human GBP-1 (Abcam, Catalog# ab131255), WARS (Thermo Fisher Scientific, Catalog# PA5-29102), STAT1 (Cell signaling, Catalog# 9172), phosphorylated STAT1 at Y701 (Cell signaling, Catalog #9167,) JAK2 (Cell signaling, Catalog#3230), and β-actin (Novus) were used. Protein expression was detected using Pierce ECL Western Blotting substrate reagent (Thermo Scientific) and ImageQuant LAS 4000 (GE Healthcare).

## Acknowledgements

This study was supported by research grants from the National Institutes of Health (to AS and MA), and Kowa Company, Ltd, Nagoya, Japan (to MA). AH thanks Alberto Santos Delgado for fruitful discussions.

## Additional information

### Funding

| Funder | Author |
| --- | --- |
| NIH Office of the Director | Amitabh Sharma |
| Kowa Company Ltd, Nagoya, Japan | Masanori Aikawa |

The funders had no role in study design, data collection and interpretation, or the decision to submit the work for publication.

### Author contributions

Arda Halu, Conceptualization, Resources, Data curation, Software, Formal analysis, Validation, Investigation, Visualization, Methodology, Writing—original draft; Jian-Guo Wang, Formal analysis, Supervision, Validation, Methodology, Writing—review and editing; Hiroshi Iwata, Resources, Methodology; Alexander Mojcher, Ana Luisa Abib, Validation; Sasha A Singh, Investigation, Methodology, Writing—review and editing; Masanori Aikawa, Amitabh Sharma, Conceptualization, Supervision, Funding acquisition, Investigation, Writing—review and editing

Author ORCIDs
Arda Halu https://orcid.org/0000-0001-6217-790X
Masanori Aikawa http://orcid.org/0000-0002-9275-2079
Amitabh Sharma http://orcid.org/0000-0003-0632-4690

Decision letter and Author response
Decision letter https://doi.org/10.7554/eLife.37059.sa1
Author response https://doi.org/10.7554/eLife.37059.sa2

## Additional files

### Supplementary files

• Supplementary file 1. Topological properties of the PPI network and co-abundance networks.

• Supplementary file 2. Overlap of co-abundance networks with large-scale binary and affinity-purification-mass spectrometry (AP-MS) based interactomes (Fisher's exact test, two-sided p-values).

• Supplementary file 3. Overlap of the top prioritized (p<0.01) proteins with inflammation, innate immune response, and coronary artery disease signatures (Fisher's exact test, two-sided p-values).

• Transparent reporting form

### Data availability

All data generated or analysed during this study are included in the manuscript and supporting files. Source data files have been provided for Figures 1 and 3.

The following dataset was generated:

| Author(s) | Year | Dataset title | Dataset URL | Database and Identifier |
|---|---|---|---|---|
| Halu A, Wang JG, Iwata H, Mojcher A, Abib AL, Singh SA, Aikawa M, Sharma A | 2020 | Context-enriched interactome powered by proteomics helps the identification of novel regulators of macrophage activation | https://www.ebi.ac.uk/pride/archive/projects/PXD020942 | PRIDE, PXD020942 |

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
