## [Decision Letter]

Thank you for submitting your article "Context-enriched interactome powered by proteomics helps the identification of novel regulators of macrophage activation" for consideration by *eLife*. Your article has been reviewed by three peer reviewers, and the evaluation has been overseen by a Reviewing Editor and Aviv Regev as the Senior Editor. The reviewers have opted to remain anonymous.

The reviewers have discussed the reviews with one another and the Reviewing Editor has drafted this decision to help you prepare a revised submission.

Summary:

The paper proposes a combination of -OMICS data using a molecular network framework. The goal is to identify new regulators of macrophage activation related with cardiovascular disease (CVD). The assessment of the results is carried out by an enrichment analysis strategy, i.e. quantifying the position in the ranking of known cardio disease related drugs (from iCTNet) and pathways. A small scale loss of function experiment is performed for two newly proposed targets.

Essential revisions:

The critical questions that have to be clarified are the following:

1) Possible biases in the network construction and associated calculations.

(1a) The inclusion of small scale experiments in the network might end up biasing the results, since these experiments target well known proteins. Similarly, the network might contain other biases, e.g. smaller distances between druggable proteins.

(1b) Other related question would be if the definition of druggable proteins, obtained from a network based method, will influence their distribution in the network?

(1c) Use of the Pearson correlation instead of the partial correlation coefficient, and the possible bias towards indirect associations.

(2) Insufficient information and doubts on the procedures.

(2a) The comparison of the combined network with its two components, i.e.

PPI and co-abundance, seems to be incomplete and the specific methodological details not sufficiently clear. What is the prediction performance of the co-abundance network alone?

(2b) It is reasonable to question if the direct union of both types of networks is the best strategy. This issue should be discussed.

(2c) The reasons of additional filtering of the results obtained with the network based approach by requesting an additional level of expression – fold changes – is unclear. How and why the resulting candidates were selected requires additional clarification.

(3) The use of shortest path distances seems too simple as a measure of associations, while other alternatives, including network kernel distances,connectivity-based association measures, assortativity, and others, are commonly used in the field.

Reviewer #1:

The authors use an integrative analysis of proteomics, transcriptomics and molecular network data in order to identify new regulators of macrophage activation, which may be of interest as candidate therapeutic targets for cardiovascular disease (CVD). The proposed methodology, which combines omics-derived co-abundance networks with literature-derived molecular interaction networks, is evaluated by assessing the enrichment of the resulting top-ranked proteins in known CVD drugs as compared to a prioritization that uses only literature-derived network information, as well as by the enrichment of the top-ranked proteins in inflammation-related pathways and signatures. Moreover, for two new top-ranked proteins, in-vitro loss-of-function experiments are used to show that these proteins have a regulatory role in pro-inflammatory signaling.

Overall, the authors' approach is logical, and although the method is very similar to previous contextualized network distance based target prioritization approaches, the obtained results in the context of macrophage activation and CVD are new and interesting. However, major comments have to be addressed regarding potential biases in the network construction resulting from the inclusion of data from targeted low-throughput experiments, the comparative evaluation of network prioritization approaches (no cross-validation is performed), and the discussion of limitations regarding false positives and negatives in the combined networks and in the experimental evaluation of candidate drug targets.

(1) In the subsection “Construction of the literature-based protein-protein interaction (PPI) network”, the authors mention that the literature-based PPI network they constructed does not only include high-quality binary interactions from high-throughput Y2H screens, but also interactions from low-throughput experiments, protein 3D structures, as well as protein complexes derived from experiments that cannot confirm direct physical interactions (e.g. Co-IP). As the authors correctly mention in the third paragraph of the Introduction, such networks are often biased towards highly studied proteins. The addition of context-specific co-abundance edges may to a certain extent address limitations of the PPI network with regard to cell-type specificity, but since the authors combine the PPI network with the co-abundance network, rather than using the co-abundance information to filter the PPI network, biases and false-positives in the original PPI network will not be removed and still influence the final analyses. Moreover, as opposed to the partial correlation coefficient, the Pearson correlation used to construct the co-abundance network will identify many indirect associations and spurious associations between two variables resulting from shared dependencies on a third confounding factor variable. For these reasons, the authors should check how the results of their approach change if low-throughput experiments and sources of evidence limited to indirect associations (rather than physical interactions) are filtered out from the PPI construction. Similarly, the influence of the correlation measure used for the construction of the co-abundance network on the final results should be investigated, e.g. by testing alternative measures like the partial correlation coefficient or the biweight midcorrelation, which is more robust against outliers, and discussing limitations arising from potential confounding variables. Finally, the authors should consider and discuss whether using the co-abundance information to filter the PPI network, rather than combining the two networks (and their erroneous edges), would provide a more reliable final network.

(2) In their comparison of network prioritization methods based on known CVD drugs from iCTNet the authors relate the predictive performance for the combined networks (PPI + co-abundance networks) to the performance for using only the PPI network via ROC curve analyses. However, they do not compare the combined networks to using the co-abundance networks only, i.e. it is not clear whether the co-abundance networks alone as data source would already outperform the purely PPI-based analysis for drug prioritization, or whether only the combination of PPI + co-abundance networks provides a superior performance. Thus, ROC curves and statistical tests for evaluating the co-abundance networks only as compared to the combined networks should be added. The statistical test used for performance comparison should also be explained in more detail: The authors mention that they use a Mann-Whitney U-Test to compare the AUROC values; however, given that only one AUROC is available for each network (the PPI and the combined network), it is not clear how the Mann-Whitney U-Test is applied in this context, since this test requires more than one value per group: Do the authors use a cross-validated AUROC computation? This would be an appropriate and more robust approach than computing a single AUROC, and would enable a statistical performance comparison, but also needs to be explained in detail in the manuscript.

Especially if the authors optimize their model (e.g. by modulating the ratio of weights, as described in the second paragraph of the subsection “Addition of macrophage derived co-abundance edges increases CVD drug target prediction performance”), a cross-validation is needed in order to prevent that the same outcome data used to optimize the prediction approach is also used to evaluate it (i.e. preventing a circular analysis with misleading performance estimates).

(3) In the Abstract, the authors write that their approach revealed "top candidates for CVD therapeutic targets"; however, in the manuscript, the authors do not investigate the druggability of these candidate proteins (GBP1 and WARS), neither experimentally nor in-silico, but only experimentally asses their regulatory role using pro-inflammatory readouts. Thus, the authors should either limit the claimed discovery to have identified/confirmed the regulatory role of the candidate proteins, or alternatively, add analyses that confirm the druggability and/or show preliminary evidence confirming that CVD-specific disease phenotypes (rather than generic inflammation readouts) in a model system can be modulated specifically via GBP1 and WARS. The authors should also mention whether existing disease gene prioritization approaches would predict the same or different candidate proteins (currently no comparison is shown).

(4) The authors should explain why they use shortest path distances for quantifying network associations rather than network kernel distances or connectivity-based association measures. The multiplicity and density of interactions that interconnect proteins may provide more evidence for a functional association than the shortest path distances alone, and it is unclear whether considering longer-range shortest path distances in the analysis still provides a significant added informative value or rather spurious associations as compared to an association measure purely based on direct connectivity. These points and potential limitations of the used association measure should be discussed.

Reviewer #2:

The authors have demonstrated an interesting approach that starts with large scale data mining and end up with validation of potential target of CVD and in particular the macrophage activities in CVD.

The experimental design and workflow of the publication is clearly laid out, and it is didactic enough to walk through the manuscript without too many hurdles.

I see no major issue with the manuscript as is and found it actually quite good examplar of systems oriented study and target prioritization excercise.

Reviewer #3:

The article is well written but the Introduction and Discussion are too lengthy. My general concerns relate to:

(1) The authors have not carefully tested for confounding factors that might underlie some if not most of the observed correlations and prediction performances. Do the drug targets show some preferential presence in the co-abundance network versus PPI network and are there any degree differences between drug targets and other proteins in the co-abundance network and/or PPI network? Do co-abundance edges preferentially connect to hubs in the PPI network? If there are biases of these kinds then a random addition of edges to the network as empirical control might not capture this and modifications like adding random edges involving proteins in the co-abundance network would represent a more relevant negative control? To which extent is the definition of your drug targets influenced by network information because a network tool, iCTNet was used? Can this lead to circularity when using these drug targets as seeds in the network proximity approach?

(2) The authors state that the drug targets are not significantly close to each other in the network, yet, when used in the benchmark of the network proximity algorithm, the performance is good. Don't they have to be close in the network for the algorithm to work? Why use the LCC sometimes and the average shortest path at other times to determine closeness? What is the negative dataset used in the benchmark and can authors plot precision-recall curves as well to provide a better overview on the precision of the predictions? What is the prediction performance of the co-abundance network alone?

(3) The reasoning for why the candidate lists need to be further improved by using expression fold change data are not clear. Is the candidate list based on the network prediction not good enough? Then this can be clearly stated. Why were the candidates GBP1 and WARS picked? Because they show the highest expression fold change among the candidates? How many other proteins show as high or higher fold change justifying that the network prediction "helped" selecting among the differentially expressed genes those that are more close to known drug targets? To my non-expert understanding of the immune system, the experiments validate the function of the candidate genes in inflammation/immune response but that was an obvious hypothesis given their up-regulation in expression? How do these experiments validate the concept of using network proximity to drug targets to find new candidates? It remains unclear how the modulation of inflammation biomarkers by the candidates relates to CVD. Contrary to statements in the discussion, I cannot find experimental validation of mechanisms.

---

## [Author Response]

Reviewer #1:[…] Overall, the authors' approach is logical, and although the method is very similar to previous contextualized network distance based target prioritization approaches, the obtained results in the context of macrophage activation and CVD are new and interesting. However, major comments have to be addressed regarding potential biases in the network construction resulting from the inclusion of data from targeted low-throughput experiments, the comparative evaluation of network prioritization approaches (no cross-validation is performed), and the discussion of limitations regarding false positives and negatives in the combined networks and in the experimental evaluation of candidate drug targets.1) In the subsection “Construction of the literature-based protein-protein interaction (PPI) network”, the authors mention that the literature-based PPI network they constructed does not only include high-quality binary interactions from high-throughput Y2H screens, but also interactions from low-throughput experiments, protein 3D structures, as well as protein complexes derived from experiments that cannot confirm direct physical interactions (e.g. Co-IP). As the authors correctly mention in the third paragraph of the Introduction, such networks are often biased towards highly studied proteins. The addition of context-specific co-abundance edges may to a certain extent address limitations of the PPI network with regard to cell-type specificity, but since the authors combine the PPI network with the co-abundance network, rather than using the co-abundance information to filter the PPI network, biases and false-positives in the original PPI network will not be removed and still influence the final analyses. Moreover, as opposed to the partial correlation coefficient, the Pearson correlation used to construct the co-abundance network will identify many indirect associations and spurious associations between two variables resulting from shared dependencies on a third confounding factor variable. For these reasons, the authors should check how the results of their approach change if low-throughput experiments and sources of evidence limited to indirect associations (rather than physical interactions) are filtered out from the PPI construction. Similarly, the influence of the correlation measure used for the construction of the co-abundance network on the final results should be investigated, e.g. by testing alternative measures like the partial correlation coefficient or the biweight midcorrelation, which is more robust against outliers, and discussing limitations arising from potential confounding variables.

We agree with the reviewer that existing biases within the PPI network as well as the potential influence of the correlation measures on the resulting co-abundance networks should be studied in more detail.

To address the former, we have systematically removed the low-throughput, literature-derived edges and the indirect co-complex edges to “de-noise” the PPI network, and measured the prediction performance by comparing AUROCs. The removal of edges from low-throughput experiments indeed resulted in an increase in the AUROC for PPI only (0.831 to 0.848). Moreover, the improvement with the addition of co-abundance edges remained the same, where AUROCs were increased proportionally (M(-): 0.862 to 0.881, M(IFNγ): 0.868 to 0.878 and M(IL-4): 0.852 to 0.870) (Figure 2—figure supplement 3B). The improvement of PPI+CoA was statistically significant under k-fold cross-validation (Figure 2—figure supplement 3C), with p = 0.0236 for M(-), p=0.0042 for M(INFγ) and p=0.0431 for M(IL-4) by paired t-test (Figure 2—figure supplement 3D). To further eliminate possible noise from the PPI network, we next removed the co-complex edges. As expected, for PPI only, the AUROC improved even further with the additional removal of protein complex edges (from 0.831 to 0.855). Moreover, the improvement with the addition of co-abundance links persisted, where AUROCs were increased proportionally (M(-): 0.862 to 0.887, M(IFNγ): 0.868 to 0.886 and M(IL-4): 0.852 to 0.879) (Figure 2—figure supplement 3B). The improvement of PPI+CoA was again statistically significant under 7-fold cross-validation (Figure 2—figure supplement 3C) with improved p-values: p = 0.0180 for M(-), p=0.0029 for M(IFNγ) and p=0.0154 for M(IL-4) (Figure 2—figure supplement 3D). Collectively, these results suggest that the systematic elimination of potentially biased edges results in an improvement in target prediction performance. In light of these results, we have now opted to run the workflow with the de-noised version of the PPI and present these results in the main figures. We have added these results in the text as shown below, and the comparison in the supplementary figures.

Results:

“Furthermore, we found that the prediction performance can be further improved by (a) the systematic “de-noising” of the PPI network by removing edges based on low-throughput experiments and co-complex interactions (see Materials and methods, and Figure 2—figure supplement 3B-D)”.

To address the second issue, we have implemented two correlation measures that are robust against outliers and control for confounding factors, i.e. biweight midcorrelation and partial correlation, respectively, to build the co-abundance networks. For the biweight midcorrelation, the M(-), M(IFNγ) and M(IL-4) networks consisted of 13,118,31,193, and 14,831 edges, respectively. When added to the PPI network, the improvement with the addition of co-abundance networks over PPI only was similar to the Pearson coefficient case (Figure 2—figure supplement 8A). The final combined rankings were also similar to those of the Pearson coefficient case, with a high level of overlap (p=9.34e-99 and p=9.00e-73 for M(IFNγ) and M(IL-4), respectively) (Figure 2—figure supplement 8C). Similarly, using partial correlation, controlling for baseline M(-) abundances, the resulting co-abundance networks had 14,307, 141,768 and 85,752 edges, respectively, for M(-), M(IFNγ) and M(IL-4). The improvement with the addition of co-abundance networks over PPI only was again similar to the Pearson coefficient case (Figure 2—figure supplement 8B). The final combined rankings were also similar to those of the Pearson coefficient case, with a high level of overlaps (p=3.64e-101 and p=1.70e-94 for M(IFNγ) and M(IL-4), respectively) (Figure 2—figure supplement 8D). These results indicate that the outlier abundance values and the indirect effect of baseline abundances do not influence the downstream analysis substantially. Despite the relatively large number of added edges, especially for partial correlation, the ROC curves and resulting combined rankings are similar to Pearson correlation coefficient. We have now included a discussion of these results in the main text as shown below, and the related figures in the supplementary figures.

Results:

“We also tested how robust the prediction performance is against changes in several points in the workflow. […] These results indicate that the outlier abundance values and the indirect effect of baseline abundances do not influence the downstream analysis substantially.”

Materials and methods:

“As additional correlation measures, we calculated biweight midcorrelation and partial correlation using the WGCNA package (1) and ppcor package (2), respectively, in R. Baseline (M(-)) abundances were controlled for in the partial correlation calculation. Resulting two-sided p-values were adjusted for multiple testing using the BH procedure, and a correlation threshold of 0.90 (p<0.01) was chosen for consistency.”

Finally, the authors should consider and discuss whether using the co-abundance information to filter the PPI network, rather than combining the two networks (and their erroneous edges), would provide a more reliable final network.

In our study, we combined the PPI network with positive correlation co-abundance edges by assuming that co-expressed proteins are likely to be functionally related (3), and that protein co-expression networks represent a complementary tool to the PPI, as suggested recently by Vella et al. (4). Furthermore, protein abundances obtained through mass spectrometry and direct protein contacts detected by crosslinking and mass-spectrometry were found to be complementary by Solis-Mezarino et al. (5). We have now added these references to the second paragraph of the Discussion.

Discussion:

“Previously, co-expression networks were suggested to represent a complementary tool to the PPI network (Vella et al., 2017), and protein abundances obtained through mass spectrometry and direct protein contacts detected by crosslinking and mass-spectrometry were found to be complementary (Solis-Mezarino and Herzog, 2017).”

Nevertheless, the filtering approach raised by the reviewer is valid and important. While we adopt a simple weighted integration method of two networks to take into account the predictive potential of each network, similar to the weighted network integration method demonstrated in the study by Valentini et al. (6), undoubtedly many alternative filtering strategies remain. Here we explored the strategy of also removing existing edges from PPI in a further attempt to de-noise the PPI, using co-abundance networks this time. As removing the noisy edges clearly showed that adding does not necessarily improve prediction performance, and conversely, removing does not necessarily decrease prediction performance, we hypothesized that removing edges could be a good strategy. Specifically, we removed the negative correlation edges in addition to adding the positive correlation ones. However, the number of edges that were already present in the PPI was limited to a few hundred. Thus, the removal of these did not result in a substantial increase in the ROCs, although there was minimal (0.01%-0.02%) improvement for M(IFNγ) and M(IL-4) in the AUROCs. In other words, the improvement caused by the removal of a few hundred edges was mostly overshadowed by the addition of many more positive correlation edges. This also goes to suggest that the main shortcoming of the PPI is incompleteness (which is thought to be a large percentage of all possible edges), rather than noise and false positives, which make up a much smaller portion. Hence we adopted our initial strategy of “adding” rather than “filtering” for this study. We have discussed this additional filtering strategy and other similar strategies that could be implemented in future studies, as shown below.

Discussion:

“While we adopted a simple weighted integration method of the PPI network and co-abundance networks to take into account the predictive potential of each network, similar to the weighted network integration method demonstrated in the study by Valentini et al. (6), many alternative filtering strategies remain. […] This suggests that the main shortcoming of the PPI is incompleteness, which is thought to be a large percentage of all possible edges, rather than noise and false positives, which make up a much smaller portion.”

2) In their comparison of network prioritization methods based on known CVD drugs from iCTNet the authors relate the predictive performance for the combined networks (PPI + co-abundance networks) to the performance for using only the PPI network via ROC curve analyses. However, they do not compare the combined networks to using the co-abundance networks only, i.e. it is not clear whether the co-abundance networks alone as data source would already outperform the purely PPI-based analysis for drug prioritization, or whether only the combination of PPI + co-abundance networks provides a superior performance. […] Especially if the authors optimize their model (e.g. by modulating the ratio of weights, as described in the second paragraph of the subsection “Addition of macrophage derived co-abundance edges increases CVD drug target prediction performance”), a cross-validation is needed in order to prevent that the same outcome data used to optimize the prediction approach is also used to evaluate it (i.e. preventing a circular analysis with misleading performance estimates).

We thank the reviewer for raising these important points regarding the prediction assessment. We agree that the ROC curves corresponding to the co-abundance networks by themselves are essential for a complete comparison. We have now performed target prioritization on these networks alone and plotted the ROC curves along with the combined networks (Figure 2—figure supplement 3A). In line with our expectation that, since co-abundance networks span a certain, biology-specific part of the entire interactome, the co-abundance networks by themselves would not yield a good performance regarding the global prioritization of proteins, the prediction performance of the co-abundance networks was lower than the combined network. This result was also confirmed in the validation datasets (Figure 2—figure supplement 5 and 7). We have added these results in the main text.

Results:

“The prediction performance of the combined networks also surpassed those of co-abundance networks alone (Figure 2—figure supplement 3A).”

The reviewer is correct that Mann-Whitney U test for the comparison of ROC curves is not the most suitable tool. We had previously used this test to compare the TPR distributions of ROC curves. We agree that cross-validation is a much better and robust approach that allows for a more intuitive statistical comparison between the curves. We have now implemented k-fold cross validation (k=7 was chosen for a reasonable test set size unless otherwise noted) for the results presented in the study, and compared the results with a paired t-test (Figure 2—figure supplements 2, 3, 5 and 7). We describe this procedure in detail in the Materials and methods section. Under cross-validation, the difference between PPI and PPI+CoA ROC curves still remain significant for the original PPI including low-throughput and co-complex interactions, as well as the “de-noised” PPI (Figure 2—figure supplement 3C-D). We have also implemented k-fold cross-validation in the optimization procedure, and selected the weight ratio that yields the highest mean AUROC across all folds for each condition (Figure 2—figure supplement 4A), and re-ran the entire workflow with the updated optimized parameters.

Materials and methods:

“We implemented k-fold (k=7 was chosen for a reasonable test set size, unless otherwise noted) cross-validation to determine the statistical significance of the difference between AUROC values, and as the datasets were partitioned into folds consistently across PPI and PPI+CoA, a paired t-test was used to compare the two cases and two-tailed p-values were reported.”

Materials and methods:

“We also implemented k-fold (k=7 was chosen for a reasonable test set size, unless otherwise noted) cross-validation in the optimization procedure, and selected the weight ratio that yields the highest mean AUROC across all folds for each condition, to be used for subsequent analyses throughout the manuscript. These values were found to be 0.4, 0.4 and 0.1 for M(-), M(IFNγ) and M(IL-4), respectively.”

3) In the Abstract, the authors write that their approach revealed "top candidates for CVD therapeutic targets"; however, in the manuscript, the authors do not investigate the druggability of these candidate proteins (GBP1 and WARS), neither experimentally nor in-silico, but only experimentally asses their regulatory role using pro-inflammatory readouts. Thus, the authors should either limit the claimed discovery to have identified/confirmed the regulatory role of the candidate proteins, or alternatively, add analyses that confirm the druggability and/or show preliminary evidence confirming that CVD-specific disease phenotypes (rather than generic inflammation readouts) in a model system can be modulated specifically via GBP1 and WARS. The authors should also mention whether existing disease gene prioritization approaches would predict the same or different candidate proteins (currently no comparison is shown).

We use the expression “therapeutic target candidate for CVD” with the assumption that molecules that arise from in silico predictions seeded with known CVD targets and found to play specific regulatory roles in pro-inflammatory macrophage activation are likely to be CVD therapeutic targets. We agree with the reviewer, however, that this assumption must be made explicit. We have thus clarified this point in the text.

Results:

“With the hypothesis that molecules that arise from in silico predictions seeded with known CVD targets and found to play specific regulatory roles in pro-inflammatory macrophage activation are likely to be CVD therapeutic targets, we next sought to investigate their structural “druggability” in silico.”

We have also modified the text to tone down this notion by refraining from the use of “CVD therapeutic targets” wherever possible, as follows:

Abstract:

“Using a network proximity-based prioritization method on the combined network, we predicted potential regulators of macrophage activation.”

Abstract:

“Integrating the novel network topology with transcriptomics and proteomics revealed top candidate drivers of inflammation.”

Introduction:

“The top-ranked candidates for regulators of macrophage activation, and hence potential CVD drug targets, also showed significant enrichment with immune system as well as cardiovascular disease related signatures.”

Results:

“Combined ranking based on network topology, gene expression and protein abundance reveals novel regulators of macrophage activation”.

Discussion:

“We interrogated the resulting combined interactome to predict and highlight new regulators of macrophage activation and potential targets of CVD”.

Discussion:

“… resulted in the network proximity-based prediction of GBP1 and WARS as potential regulators of pro-inflammatory macrophage activation.”

Furthermore, we explored the structural druggability of the top five candidates in silico, using the DrugEBIlity database. The top five candidates of pro-inflammatory macrophage activation all contained binding sites suitable for small molecules, with at least 50% tractable structures and 20% druggable structures (Figure 5—figure supplement 2A-B). Among these, WARS and GBP1 were the two candidates with the highest percentage of druggable structures (Figure 5—figure supplement 2B). These results are added in the main text.

Results:

“The top five candidates of pro-inflammatory macrophage activation all contained binding sites suitable for small molecules, with at least 50% tractable structures and 20% druggable structures (see Materials and methods, Figure 5—figure supplement 2A-B). Among these, WARS and GBP1 were the two candidates with the highest percentage of druggable structures (Figure 5—figure supplement 2B).”

Materials and methods:

“To assess the potential of candidates of interest to be drug targets, we used the DrugEBIlity database (https://www.ebi.ac.uk/chembl/drugebility/) (version 3.0), which predicts the structural druggability of a molecule by how suitable its binding sites are for small molecules under the Lipinski's Rule of 5 requiring at most 10 hydrogen bond acceptors, at most 5 hydrogen bond donors, and a molecular weight 500 Da or less. […] “Ensemble druggability” is the strictest criterion where the average of druggability score is calculated under different machine learning models.”

Regarding the comparison with other disease gene prioritization approaches, we describe our results below where a similar point is raised by the reviewer.

4) The authors should explain why they use shortest path distances for quantifying network associations rather than network kernel distances or connectivity-based association measures. The multiplicity and density of interactions that interconnect proteins may provide more evidence for a functional association than the shortest path distances alone, and it is unclear whether considering longer-range shortest path distances in the analysis still provides a significant added informative value or rather spurious associations as compared to an association measure purely based on direct connectivity. These points and potential limitations of the used association measure should be discussed.

For our study, we chose shortest path related measures as a proxy of association with drug targets, as they are well studied and widely used within the field for disease gene and drug target candidate prioritization. Nevertheless, we agree with the reviewer that they should be compared to global dynamical association measures, as well as other types of network-based distance measures. To explore how sensitive prediction performance and the resulting candidates are to untraversed longer paths, we have considered dynamical prioritization methods such as random walk with restarts (RWR) and its degree-aware version (DADA), as well as other distance-based measures such as the kernel distance. Relying on a random walker, RWR/DADA takes into account all possible paths, including many longer-range paths between a candidate and a target (seed), whereas Kernel distance penalizes paths based on their length using an exponential function. Both of these measures yielded similar results to average shortest path in terms of the AUC, and the improvement with the addition of CoA edges persisted (Figure 2—figure supplement 9A and 9D). Moreover, both the top prioritization and the top combined-ranked proteins significantly overlapped between our method and these additional measures (Figure 2—figure supplement 9B, 9C, 9E and 9F). These results indicate that the improvement of target prediction with context-specific information from co-abundance networks is independent of the prioritization metric used. We have included the related results in the Results section, and in the Materials and methods, as shown below. While the results are similar, we also describe some limitations of the shortest path based approaches in the Discussion section.

Results:

“Second, to explore how sensitive prediction performance is to untraversed longer-range paths between candidates and drug targets, we have considered global association measures and other types of network-based distance measures. […] Both types of measures yielded similar results to average shortest path in terms of the AUROC, and the improvement with the addition of CoA edges persisted (Figure 2—figure supplement 9).”

Results:

“Finally, the top prioritized and the top combined-ranked proteins showed good agreement with the rankings found through the alternative correlation measures and prioritization methods discussed above, showing significant overlap (Figure 2—figure supplements 8C-D, 9B-C and 9E-F).”

Materials and methods:

“As additional prioritization measures, we implemented RWR and DADA as described in (7), and Kernel distance *dk(c)* per candidate *c* was defined asdk(c)=−ln∑s∈Se−(dcs+1)|S|,where *d_cs_* is the shortest network distance between candidate gene *c* and seed gene *s* and *S* is the set of all seed genes.”

Discussion:

“Considering potential limitations of shortest path based prioritization methods such as ignoring biologically meaningful alternative longer-range paths between candidates and drug targets, we also ran our workflow on global dynamical prioritization measures, which displayed similar results, suggesting that the improvement of target prediction with context-specific information from co-abundance networks is independent of the prioritization metric used.”

Note: Other similar distance based measures such as “closest” and “center” distances, which have been used in determining the proximity between two gene sets (8), have limited practicality as prioritization measures, as these depend on only one shortest distance per candidate, resulting in lots of redundancies in the rankings (i.e. there will be hundreds of candidates that have the same “closest” or “center” distance value of 1 or 2). We have therefore not included these in the list of measures to be compared against.

Reviewer #3:The article is well written but the Introduction and Discussion are too lengthy. My general concerns relate to:1) The authors have not carefully tested for confounding factors that might underlie some if not most of the observed correlations and prediction performances. Do the drug targets show some preferential presence in the co-abundance network versus PPI network and are there any degree differences between drug targets and other proteins in the co-abundance network and/or PPI network? Do co-abundance edges preferentially connect to hubs in the PPI network? If there are biases of these kinds then a random addition of edges to the network as empirical control might not capture this and modifications like adding random edges involving proteins in the co-abundance network would represent a more relevant negative control? To which extent is the definition of your drug targets influenced by network information because a network tool, iCTNet was used? Can this lead to circularity when using these drug targets as seeds in the network proximity approach?

We thank the reviewer for raising these important points about possible confounding factors. It is indeed crucial to determine these potential relationships. We have addressed these concerns one by one:

The proportions of CVD drug targets to non-drug targets in the PPI and co-abundance network were very similar to each other (0.0018 versus 0.0017, respectively), suggesting that there is no preferential presence of drug targets in these networks. We have now added this result in the text.

Results:

“The proportions of CVD drug targets to non-drug targets in the PPI and co-abundance network were very similar to each other (0.0018 versus 0.0017, respectively), suggesting that there is no preferential presence of drug targets in either network over the other one.”

Regarding biases related to degrees: While the degrees of drug targets in the co-abundance networks are not significantly different from the degrees of other proteins in the co-abundance networks, the degrees of drug targets in the PPI network are significantly higher than the degrees of other proteins in the PPI network (Figure 2—figure supplement 1B). Furthermore, measuring the tendency of co-abundance edges to connect to hubs in the PPI network, we found that proteins connected by CoA edges (i.e. CoA network nodes) have significantly higher degrees in the PPI network than the other nodes in the PPI network (Figure 2—figure supplement 1C). Collectively, these results point at a degree bias in the way drug targets and co-abundance edges are connected in the PPI network. We have summarized these results, as shown below.

Results:

“In terms of connectivity, the degrees of drug targets were significantly higher than the degrees of other proteins in the PPI network, while there was no significant degree difference between drug targets and other proteins in the co-abundance networks (Figure 2—figure supplement 1B). […] Taken together, these results point at a degree bias in the way drug targets and co-abundance edges are connected in the PPI network.”

To take into account the bias in the connectivity between the co-abundance and PPI networks while prioritizing the entire interactome, we changed the randomization procedure to incorporate degree-preserving randomization. This approach effectively permits us to add randomly selected edges that connected proteins with similar degrees to the proteins in the co-abundance networks. We re-ran the entire workflow with this updated procedure. The degree-preserving random edge addition procedure is now described in the Results and Materials and methods sections, and we have updated Figures 1D and 2D to reflect this change, as well as revised the legend for Figure 2A to refer the reader to the above results.

Results:

“To control for the degree bias in the connectivity between co-abundance and PPI networks (Figure 2—figure supplement 1B-C), we implemented degree-preserving randomization, which ensures the addition of randomly selected edges that connect proteins with similar degrees to the proteins in the co-abundance networks (see Materials and methods).”

Materials and methods:

“For the addition of random edges used as negative control in the prediction performance assessment, we adopted a degree-preserving randomization strategy. […] Edges were then established between pairs of proteins uniformly randomly picked from their respective degree bin representing a pool of similar degree proteins.”

Regarding the circularity concern: iCTNet itself is not a network-based method to identify disease genes or drug targets, but rather a heterogeneous network representation of the publicly available information regarding various independent types of interactions such as gene-tissue, drug-disease, miRNA-gene interactions, etc. (9). The drug target information used in our manuscript employs the set of drug-gene and drug-disease interactions in the iCTNet database, which are obtained from the Comparative Toxigenomics Database (CTD) and the DrugBank database. While iCTNet also has protein-protein interactions, they are not used in our study, and in any case they are independent from the drug-gene and drug-disease interactions. Due to this independence between sources and the lack of involvement of protein interactions in the drug-gene and drug-disease, there is no prospect of circularity. We have now clarified this point in the Materials and methods section.

Materials and methods:

“While the iCTNet database also has protein-protein interactions, they were not used in our study to avoid circularity.”

2) The authors state that the drug targets are not significantly close to each other in the network, yet, when used in the benchmark of the network proximity algorithm, the performance is good. Don't they have to be close in the network for the algorithm to work? Why use the LCC sometimes and the average shortest path at other times to determine closeness? What is the negative dataset used in the benchmark and can authors plot precision-recall curves as well to provide a better overview on the precision of the predictions? What is the prediction performance of the co-abundance network alone?

We would like to distinguish between the closeness of drug targets to each other and the closeness of target candidates to drug targets. The exploration of the former is related to the question of how clustered together versus dispersed they are around the interactome – or how complete or fragmented the CVD drug target information is in the interactome. The LCC analysis is related to this aspect. As such, the closeness of drug targets to each other does not play a role in the comparison of prediction performance. The latter case, i.e. the closeness of targets candidates to drug targets, is what determines the candidate’s ranking and the resulting ROC curves and prediction performance comparisons. When determining the closeness, always the shortest paths are used. The negative control used to determine the statistical significance of the improvement is the degree-preserved randomly added edges to enable a fair comparison with the co-abundance nodes. We have now expanded upon the above where applicable in the text as well as the sections related to Figure 2A and Figure 2—figure supplement 1A.

Results:

“The co-abundance networks spanned a certain portion of the entire PPI network with their edges concentrated on certain regions, while the CVD drug targets were dispersed around the combined PPI, forming a disconnected subnetwork (Figure 2A and Figure 2—figure supplement 1A), suggesting the fragmented nature of the CVD drug target information in the interactome.”

Figure 2A legend:

“The entire literature curated PPI network with co-abundance edges from all three stimulation conditions, providing a global view of the distribution and connectivity of co-abundance edges and drug targets (Figure 2—figure supplement 1 for a quantification of related degree distributions and largest connected component (LCC) sizes). A force-directed layout algorithm was used to visualize the networks.”

While we recognize that precision-recall curves can be more useful in imbalanced datasets where positives are rare, we chose to present ROC curves to allow for a consistent comparison across different validation datasets, regardless of the ratio of positives (known drug targets) to negatives (non-drug targets, i.e., drug target candidates) in each dataset. This enabled, for instance, the comparison between PPI+CoA and CoA only networks within a dataset, as well as comparisons across different datasets. Nevertheless, for interested readers, we have followed the reviewer’s suggestion and plotted additional precision-recall curves. These are now made available as the right-hand side plots of Figure 2—figure supplements 2, 5 and 7, and the bottom row in Figure 2—figure supplement 6.

Methods:

“In addition to ROC curves, we additionally plotted precision-recall curves as they can provide additional information for imbalanced datasets where positives are rare.”

Finally, we agree that the ROC curves corresponding to the co-abundance networks by themselves are essential for a complete comparison. We have now performed target prioritization on these networks alone and plotted the ROC curves along with the combined networks (Figure 2—figure supplement 3A). In line with our expectation that, since co-abundance networks span a certain, biology-specific part of the entire interactome, the co-abundance networks by themselves would not yield a good performance regarding the global prioritization of proteins, the prediction performance of the co-abundance networks was lower than the combined network. This result was also confirmed in the validation datasets (Figure 2—figure supplements 5 and 7). We have added these results in the main text.

Results:

“The prediction performance of the combined networks also surpassed those of co-abundance networks alone (Figure 2—figure supplement 3A).”

3) The reasoning for why the candidate lists need to be further improved by using expression fold change data are not clear. Is the candidate list based on the network prediction not good enough? Then this can be clearly stated. Why were the candidates GBP1 and WARS picked? Because they show the highest expression fold change among the candidates? How many other proteins show as high or higher fold change justifying that the network prediction "helped" selecting among the differentially expressed genes those that are more close to known drug targets?

We agree that the rationale behind the combined ranking should be further clarified and the contributions of each type of ranking should be detailed. We appreciate this important advice.

After showing that the global ranking of all proteins in the interactome is improved with the addition of co-abundance links, we sought to focus on a much smaller subspace of highly expressed candidates for further validation. This is not to say that proteins highly ranked by prioritization ranking only do not have the potential to be viable drug targets/regulators of macrophage activation. Rather, we chose to add the omics to detect the strongest signals that are also close to drug targets. GBP1 and WARS are indeed strongly induced in terms of fold change. Our final ranking does not, however, reflect the expression ranking directly. To address these questions more concretely, we have therefore conducted analyses to present a “micro” and “macro” view of the ranking process.

As for the “micro” view, GBP5 and CXCL10 were both higher than GBP1 in terms of expression fold change, but their closeness to drug targets were both lower than GBP1 due to their very low ranking (12978 and 10804, respectively) in the prioritization, therefore they were deprioritized relative to GBP1. Similarly, GBP4, CXCL9, CCL8, TNFAIP6, IFIT2, EDN1, IFIT3, IRF1 all had higher expression than WARS, but their closeness to drug targets were lower than WARS, pushing them lower in the ranks due to their low rankings (12981, 11039, 10649, 10542, 12495, 3853, 5518 and 4007, respectively) in the prioritization. Therefore, these also were deprioritized relative to WARS.

For the “macro” view, we focused on the three rankings and their relationship, as well as their relative contribution to the combined ranking. To quantify each omics source’s relative contribution to the final ranking, we plotted the ranks of each protein in the final list, according to their combined rank and the three constituent rankings. The combined ranking was positively correlated (as expected) with all separate rankings. While the combined ranking was slightly more driven by the gene expression and protein abundance (Spearman’s Rho=0.64 (p=3.10e-06) for expression FC and Spearman’s Rho=0.67 (p=1.02e-06) for abundance, respectively) the network prioritization ranking was close to them for M(IFNγ) (Spearman’s Rho=0.31 (p=0.041)) (Figure 4—figure supplement 2A) and on par with them for M(IL-4) (Spearman’s Rho=0.60 (p=1.34e-03)) (Figure 4—figure supplement 2B). Moreover, each separate ranking was orthogonal to each other, showing no correlation among themselves (Spearman’s Rho ranging between -0.074 and 0.097 (p>0.05)) (Figure 4—figure supplement 3). Taken together, these results suggested that network-based prioritization ranking contributes in a non-trivial way to the combined ranking, and that each ranking carries its own information, contributing uniquely to the combined ranking. We have now added these results and incorporated the related discussion in the main text. We have also expanded the description of and the rationale behind the combined filtering/ranking approach.

Results:

“In a similar vein, we sought to integrate our protein abundance data with previously published gene expression data from human macrophages (10) to refine our list of prioritized proteins and focus on a much smaller subspace of highly expressed candidates for further in vitro validation.”

Results:

“The filtering step was used to intersect the network-based prioritization ranking with the highly expressed molecules from -omics data to detect the strongest signals that were also close to drug targets. The ranking scheme considered (a) the network closeness to drug targets, (b) the relative protein abundance difference with respect to baseline M(-) over all time points, and (c) the gene expression fold change with respect to baseline M(-) from human macrophage transcriptome data, and calculated a combined score based on these three criteria (Figure 4A and Figure 4—figure supplement 1, see Materials and methods)”.

Results:

“To quantify each constituent ranking’s relative contribution to the final ranking, we investigated the correlations between them. […] Together, these results suggest that network-based prioritization ranking contributes in a non-trivial way to the combined ranking, and that each ranking carries its own information, contributing uniquely to the combined ranking.”

To my non-expert understanding of the immune system, the experiments validate the function of the candidate genes in inflammation/immune response but that was an obvious hypothesis given their up-regulation in expression? How do these experiments validate the concept of using network proximity to drug targets to find new candidates? It remains unclear how the modulation of inflammation biomarkers by the candidates relates to CVD. Contrary to statements in the discussion, I cannot find experimental validation of mechanisms.

The reviewer is correct that the in vitro experiments were aimed at establishing the regulatory role of GBP1 and WARS in pro-inflammatory macrophage activation. Although systems approaches facilitate target discovery, we are aware that increased or decreased genes or proteins identified by unbiased screening may not necessarily play causal roles. The expression levels of certain proteins, including GBP1 and WARS, increase during activation of human macrophages, as gauged by induction of pro-inflammatory molecules, but these molecules may not contribute to this phenotypic switch. We thus performed loss-of-function experiments using siRNA silencing for GBP1 and WARS. In addition, such validation experiments are also critical to examine whether our systems approach can provide biologically meaningful results. To do this, we chose GBP1 and WARS as examples.

Inflammation plays important roles in the development and complications of atherosclerotic diseases (e.g., myocardial infarction). Despite extensive studies on vascular diseases and the development of various potent drugs, particularly lipid lowering statins, many patients still suffer with complications of inflammatory vascular diseases. Such residual risk is a global health burden. This clinical unmet need has driven active target discovery efforts to explore new mechanisms for inflammation beyond modifiable risk factors (e.g., dyslipidemia) and translate new findings into the clinic. The present study thus aimed to explore molecules that promote pro-inflammatory activation of macrophages using unbiased proteomics. To select promising targets, we used computational prediction that linked GBP1 and WARS with cardiovascular diseases. in vitro validation experiments, using the human macrophage-like cell line THP-1 and human primary macrophages, demonstrated that GBP1 and WARS indeed regulate macrophage activation.

We agree with the reviewer, however, that these points should be made clear in the text, and the discussion regarding potential therapeutic targets should be broadened to include other possible inflammatory diseases. We have now revised the text in this regard.

Results:

“Although a systems approach facilitates target discovery, increased or decreased genes or proteins identified by unbiased omics screening may not necessarily play causal roles. […] Thus, to provide mechanistic insights about these induced molecules, as well as to validate our systems approach, we performed in vitro loss-of-function experiments.”

We also agree that we should limit the use of such an expression as “therapeutic targets.” We have also modified the text to refrain from the use of “CVD therapeutic targets” wherever possible, as shown below:

Abstract:

“Using a network proximity-based prioritization method on the combined network, we predicted potential regulators of macrophage activation.”

Abstract:

“Integrating the novel network topology with transcriptomics and proteomics revealed top candidate drivers of inflammation.”

Introduction:

“The top-ranked candidates for regulators of macrophage activation, and hence potential CVD drug targets, also showed significant enrichment with immune system as well as cardiovascular disease related signatures.”

Results:

“Combined ranking based on network topology, gene expression and protein abundance reveals novel regulators of macrophage activation”.

Discussion:

“We interrogated the resulting combined interactome to predict and highlight new regulators of macrophage activation and potential targets of CVD”.

Discussion:

“… resulted in the network proximity-based prediction of GBP1 and WARS as potential regulators of pro-inflammatory macrophage activation.”

In addition, we have explored the structural druggability of the top five candidates in silico, using the DrugEBIlity database. The top five candidates of pro-inflammatory macrophage activation all contained binding sites suitable for small molecules, with at least 50% tractable structures and 20% druggable structures (Figure 5—figure supplement 2A-B). Among these, WARS and GBP1 were the two candidates with the highest percentage of druggable structures (Figure 5 —figure supplement 2B). These results are added in the main text.

Results:

“With the hypothesis that molecules that arise from in silico predictions seeded with known CVD targets and found to play specific regulatory roles in pro-inflammatory macrophage activation are likely to be CVD therapeutic targets, we next sought to investigate their structural “druggability” in silico. The top five candidates of pro-inflammatory macrophage activation all contained binding sites suitable for small molecules, with at least 50% tractable structures and 20% druggable structures (see Materials and methods, Figure 5—figure supplement 2A-B). Among these, WARS and GBP1 were the two candidates with the highest percentage of druggable structures (Figure 5—figure supplement 2B).”

Methods:

“To assess the potential of candidates of interest to be drug targets, we used the DrugEBIlity database (https://www.ebi.ac.uk/chembl/drugebility/) (version 3.0), which predicts the structural druggability of a molecule by how suitable its binding sites are for small molecules under the Lipinski's Rule of 5 requiring at most 10 hydrogen bond acceptors, at most 5 hydrogen bond donors, and a molecular weight 500 Da or less. […] “Ensemble druggability” is the strictest criterion where the average of druggability score is calculated under different machine learning models.”

References:

1) Langfelder,P. and Horvath,S. (2012) Fast R Functions for Robust Correlations and Hierarchical Clustering. J. Stat. Softw., 46, 1–17.

2) Kim,S. (2015) ppcor: An R Package for a Fast Calculation to Semi-partial Correlation Coefficients. Commun. Stat. Appl. methods, 22, 665–674.

3) Kustatscher,G., Grabowski,P. and Rappsilber,J. (2017) Pervasive coexpression of spatially proximal genes is buffered at the protein level. Mol. Syst. Biol., 13, 937.

4) Vella,D., Zoppis,I., Mauri,G., Mauri,P. and Di Silvestre,D. (2017) From protein-protein interactions to protein co-expression networks: a new perspective to evaluate large-scale proteomic data. EURASIP J. Bioinform. Syst. Biol., 2017, 6.

5) Solis-Mezarino,V. and Herzog,F. (2017) compleXView: a server for the interpretation of protein abundance and connectivity information to identify protein complexes. Nucleic Acids Res., 45, W276–W284.

6) Valentini,G., Paccanaro,A., Caniza,H., Romero,A.E. and Re,M. (2014) An extensive analysis of disease-gene associations using network integration and fast kernel-based gene prioritization methods. Artif. Intell. Med., 61, 63–78.

7) Erten,S., Bebek,G., Ewing,R.M. and Koyutürk,M. (2011) DADA: Degree-Aware Algorithms for Network-Based Disease Gene Prioritization. BioData Min., 4, 19.

8) Guney,E., Menche,J., Vidal,M. and Barábasi,A.-L. (2016) Network-based in silico drug efficacy screening. Nat. Commun., 7, 10331.

9) Wang,L., Himmelstein,D.S., Santaniello,A., Parvin,M. and Baranzini,S.E. (2015) iCTNet2: integrating heterogeneous biological interactions to understand complex traits. F1000Research, 4.

10) Xue,J., Schmidt,S. V, Sander,J., Draffehn,A., Krebs,W., Quester,I., De Nardo,D., Gohel,T.D., Emde,M., Schmidleithner,L., et al. (2014) Transcriptome-based network analysis reveals a spectrum model of human macrophage activation. Immunity, 40, 274–288.